# ADVANTAGE-AWARE POLICY OPTIMIZATION FOR OFFLINE REINFORCEMENT LEARNING

## ABSTRACT

Offline Reinforcement Learning (RL) endeavors to leverage offline datasets to craft effective agent policy without online interaction, which imposes proper conservative constraints with the support of behavior policies to tackle the Out-Of-Distribution (OOD) problem. However, existing works often suffer from the constraint conflict issue when offline datasets are collected from multiple behavior policies, *i.e.,* different behavior policies may exhibit inconsistent actions with distinct returns across the state space. To remedy this issue, recent Advantage-Weighted (AW) methods prioritize samples with high advantage values for agent training while inevitably leading to overfitting on these samples. In this paper, we introduce a novel Advantage-Aware Policy Optimization (A2PO) method to explicitly construct advantage-aware policy constraints for offline learning under the mixed-quality datasets. Specifically, A2PO employs a Conditional Variational Auto-Encoder (CVAE) to disentangle the action distributions of intertwined behavior policies by modeling the advantage values of all training data as conditional variables. Then the agent can follow such disentangled action distribution constraints to optimize the advantage-aware policy towards high advantage values. Extensive experiments conducted on both the single-quality and mixed-quality datasets of the D4RL benchmark demonstrate that A2PO yields results superior to state-of-the-art counterparts. Our code will be made publicly available.

## 1 INTRODUCTION

Offline Reinforcement Learning (RL) (Fujimoto et al., 2019; Chen et al., 2020) aims to learn effective control policies from pre-collected datasets without online exploration, and has witnessed its unprecedented success in various real-world applications, including robot manipulation (Xiao et al., 2022; Lyu et al., 2022), recommendation system (Zhang et al., 2022; Sakhi et al., 2023), *etc*. A formidable challenge of offline RL lies in the Out-Of-Distribution (OOD) problem (Levine et al., 2020), involving the distribution shift between data induced by the learned policy and data collected by the behavior policy. Consequently, the direct application of conventional online RL methods inevitably exhibits extrapolation error (Prudencio et al., 2023), where the unseen state-action pairs are erroneously estimated. To tackle this OOD problem, offline RL methods attempt to impose proper conservatism on the learning agent within the distribution of the dataset, such as restricting the learned policy with a regularization term (Kumar et al., 2019; Fujimoto & Gu, 2021) or penalizing the value overestimation of OOD actions (Kumar et al., 2020; Kostrikov et al., 2021).

Despite the promising results achieved, offline RL often encounters the constraint conflict issue when dealing with the mixed-quality dataset (Chen et al., 2022; Singh et al., 2022; Gao et al., 2023; Chebotar et al., 2023). Specifically, when training data are collected from multiple behavior policies with distinct returns, existing works still treat each sample constraint equally with no regard for the differences in data quality. This oversight can lead to conflict value estimation and further suboptimal results. To resolve this concern, the Advantage-Weighted (AW) methods employ weighted sampling to prioritize training transitions with high advantage values from the offline dataset (Chen et al., 2022; Tian et al., 2023; Zhuang et al., 2023). However, we argue that these AW methods implicitly reduce the diverse behavior policies associated with the offline dataset into a narrow one from the viewpoint of the dataset redistribution. As a result, this redistribution operation of AW may exclude a substantial number of crucial transitions during training, thus impeding the advantage estimation for the effective state-action space. To exemplify the advantage estimation problem in AW,

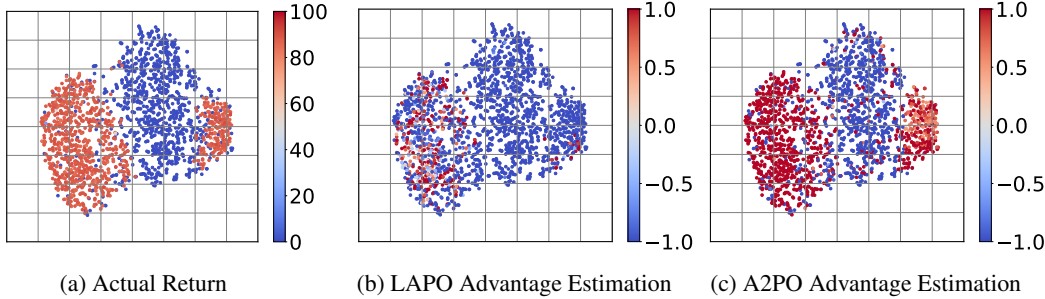

| (a) Actual Return | (b) LAPO Advantage Estimation | (c) A2PO Advantage Estimation |

Figure 1: Comparison of our proposed A2PO method and the state-of-the-art AW method (LAPO) in advantage estimation for the mixed-quality offline dataset (*halfcheetah-random-expert-v2*). Each data point represents an initial state-action pair in the offline dataset after applying PCA, while varying shades of color indicate the magnitude of the actual return or advantage value.

we conduct a didactic experiment on the state-of-the-art AW method, LAPO (Chen et al., 2022), as shown in Figure 1. The results demonstrate that LAPO only accurately estimates the advantage value for a small subset of the high-return state-action pairs (left part of Figure 1b), while consistently underestimating the advantage value of numerous effective state-action pairs (right part of Figure 1b). These errors in advantage estimation can further lead to unreliable policy optimization.

In this paper, we propose *Advantage-Aware Policy Optimization*, abbreviated as A2PO, to explicitly learn the advantage-aware policy with disentangled behavior policies from the mixed-quality offline dataset. Unlike previous AW methods devoted to dataset redistribution while overfitting on high-advantage data, the proposed A2PO directly conditions the agent policy on the advantage values of all training data without any prior preference. Technically, A2PO comprises two alternating stages, *behavior policy disentangling* and *agent policy optimization*. The former stage introduces a Conditional Variational Auto-Encoder (CVAE) (Sohn et al., 2015) to disentangle different behavior policies into separate action distributions by modeling the advantage values of collected state-action pairs as conditioned variables. The latter stage further imposes an explicit advantage-aware policy constraint on the training agent within the support of disentangled action distributions. Combining policy evaluation and improvement with such advantage-aware constraint, A2PO can perform a more effective advantage estimation, as illustrated in Figure 1c, to further optimize the agent toward high advantage values to obtain the effective decision-making policy.

To sum up, our main contribution is the first dedicated attempt towards advantage-aware policy optimization to alleviate the constraint conflict issue under the mixed-quality offline dataset. The proposed A2PO can achieve advantage-aware policy constraint derived from different behavior policies, where a customized CVAE is employed to infer diverse action distributions associated with the behavior policies by modeling advantage values as conditional variables. Extensive experiments conducted on the D4RL benchmark (Fu et al., 2020), including both single-quality and mixed-quality datasets, demonstrate that the proposed A2PO method yields significantly superior performance to the state-of-the-art offline RL baselines, as well as the advantage-weighted competitors.

## 2 RELATED WORKS

**Offline RL** can be broadly classified into three categories: policy constraint (Fujimoto et al., 2019; Vuong et al., 2022), value regularization (Ghasemipour et al., 2022; Hong et al., 2022), and model-based methods (Kidambi et al., 2020; Yang et al., 2023). Policy constraint methods attempt to impose constraints on the learned policy to be close to the behavior policy (Kumar et al., 2019). Previous studies directly introduce the explicit policy constraint for agent learning, such as behavior cloning (Fujimoto & Gu, 2021), maximum mean discrepancy (Kumar et al., 2019), or maximum likelihood estimation (Wu et al., 2022). In contrast, recent efforts mainly focus on realizing the policy constraints in an implicit way (Peng et al., 2019; Yang et al., 2021; Nair et al., 2020; Siegel et al., 2020), which approximates the formal optimal policy derived from KL-divergence constraint. On the other hand, value regularization methods make constraints on the value function to alleviate

the overestimation of OOD action. Kumar et al. (2020) approximate the lower bound of the value function by incorporating a conservative penalty term encouraging conservative action selection. Similarly, Kostrikov et al. (2021) adopt expectile regression to perform conservative estimation of the value function. To mitigate the overpessimism problem in the value regularization methods, Lyu et al. (2022) construct a mildly conservative Bellman operator for value network training. Model-based methods construct the environment dynamics to estimate state-action uncertainty for OOD penalty (Yu et al., 2020; 2021). However, in the context of offline RL with the mixed-quality dataset, all these methods treat each sample constraint equally without considering data quality, thereby resulting in conflict value estimation and further suboptimal learning outcomes.

**Advantage-weighted offline RL method** employs weighted sampling to prioritize training transitions with high advantage values from the offline dataset. To enhance sample efficiency, Peng et al. (2019) introduce an advantage-weighted maximum likelihood loss by directly calculating advantage values via trajectory return. Nair et al. (2020) further use the critic network to estimate advantage values for advantage-weighted policy training. Recently, AW methods have also been well studied in addressing the constraint conflict issue that arises from the mixed-quality dataset (Chen et al., 2022; Zhuang et al., 2023; Peng et al., 2023). Several studies present advantage-weighted behavior cloning as a direct objective function (Zhuang et al., 2023) or an explicit policy constraint (Fujimoto & Gu, 2021). Chen et al. (2022) propose the Latent Advantage-Weighted Policy Optimization (LAPO) framework, which employs an advantage-weighted loss to train CVAE for generating high-advantage actions based on the state condition. However, this AW mechanism inevitably suffers from overfitting to specific high-advantage samples. Meanwhile, return-conditioned supervised learning (Brandfonbrener et al., 2022) learns the action distribution with explicit trajectory return signals. In contrast, our A2PO directly conditions the agent policy on both the state and the estimated advantage value, enabling effective utilization of all samples with varying quality.

## 3 PRELIMINARIES

We formalize the RL task as a Markov Decision Process (MDP) (Puterman, 2014) defined by a tuple $\mathcal{M} = \langle \mathcal{S}, \mathcal{A}, P, r, \gamma, \rho_0 \rangle$, where $\mathcal{S}$ represents the state space, $\mathcal{A}$ represents the action space, $P : \mathcal{S} \times \mathcal{A} \times \mathcal{S} \to [0,1]$ denotes the environment dynamics, $r : \mathcal{S} \times \mathcal{A} \to \mathbb{R}$ denotes the reward function, $\gamma \in (0,1]$ is the discount factor, and $\rho_0$ is the initial state distribution. At each time step $t$, the agent observes the state $s_t \in \mathcal{S}$ and selects an action $a_t \in \mathcal{A}$ according to its policy $\pi$. This action leads to a transition to the next state $s_{t+1}$ based on the dynamics distribution $P$. Additionally, the agent receives a reward signal $r_t$. The goal of RL is to learn an optimal policy $\pi^*$ that maximizes the expected return: $\pi^* = \arg\max_\pi \mathbb{E}_\pi \left[ \sum_{k=0}^\infty \gamma^k r_{t+k} \right]$. In offline RL, the agent can only learn from an offline dataset without online interaction with the environment. In the single-quality settings, the offline dataset $\mathcal{D} = \{(s_t, a_t, r_t, s_{t+1}) \mid t = 1, \cdots, N\}$ with $N$ transitions is collected by only one behavior policy $\pi_\beta$. In the mixed-quality settings, the offline dataset $\mathcal{D} = \bigcup_i \{(s_{i,t}, a_{i,t}, r_{i,t}, s_{i,t+1}) \mid t = 1, \cdots, N\}$ is collected by multiple behavior policies $\{\pi_{\beta_i}\}_{i=1}^M$.

In the context of RL, we evaluate the learned policy $\pi$ by the state-action value function $Q^\pi(s, a) = \mathbb{E}_\pi \left[ \sum_{t=0}^\infty \gamma^t r(s_t, a_t) \mid s_0 = s, a_0 = a \right]$. The value function is defined as $V^\pi(s) = \mathbb{E}_{a\sim\pi} \left[ Q^\pi(s, a) \right]$, while the advantage function is defined as $A^\pi(s, a) = Q^\pi(s, a) - V^\pi(s)$. For continuous control, our A2PO implementation uses the TD3 algorithm (Fujimoto et al., 2018) based on the actor-critic framework as a basic backbone for its robust performance. The actor network $\pi_\omega$, known as the learned policy, is parameterized by $\omega$, while the critic networks consist of the Q-network $Q_\theta$ parameterized by $\theta$ and the V-network $V_\phi$ parameterized by $\phi$. The actor-critic framework involves two steps: policy evaluation and policy improvement. During policy evaluation phase, the Q-network $Q_\theta$ is optimized by following temporal-difference (TD) loss ((Sutton & Barto, 2018)):

$$\mathcal{L}_Q(\theta) = \mathbb{E}_{(s,a,r,s')\sim\mathcal{D}, a'\sim\pi_{\hat\omega}(s')} \left[ Q_\theta(s, a) - \left( r(s, a) + \gamma Q_{\hat\theta}(s', a') \right) \right]^2, \tag{1}$$

where $\hat\theta$ and $\hat\omega$ are the parameters of the target networks that are regularly updated by online parameters $\theta$ and $\omega$ to maintain learning stability. The V-network $V_\phi$ can also be optimized by the similar TD loss. For policy improvement in continuous control, the actor network $\pi_\omega$ can be optimized by the deterministic policy gradient loss (Silver et al., 2014; Schulman et al., 2017):

$$\mathcal{L}_\pi(\omega) = \mathbb{E}_{s\sim\mathcal{D}} \left[ -Q_\theta(s, \pi_\omega(s)) \right]. \tag{2}$$

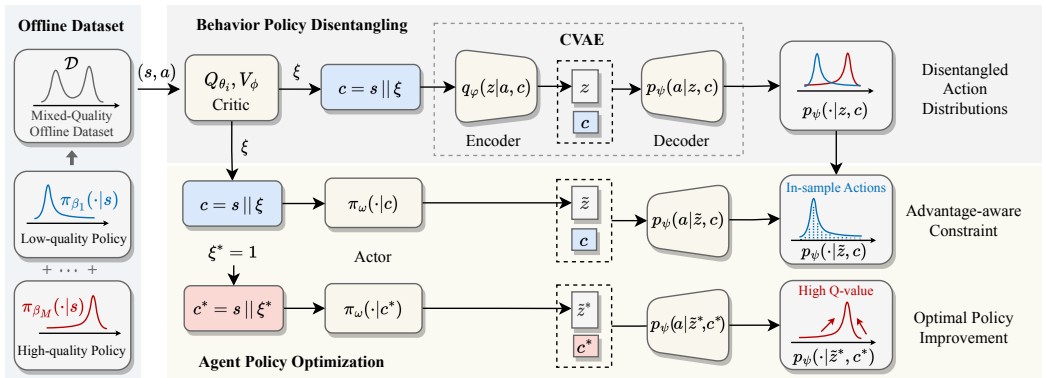

Figure 2: An illustrative diagram of the Advantage-Aware Policy Optimization (A2PO) method.

Note that offline RL will impose conservative constraints on the optimization losses to tackle the OOD problem. Moreover, the performance of the final learned policy $\pi_\omega$ highly depends on the quality of the offline dataset $\mathcal{D}$ associated with the behavior policies $\{\pi_{\beta_i}\}$.

## 4 METHODOLOGY

In this section, we provide details of our proposed A2PO approach, consisting of two key components: *behavior policy disentangling* and *agent policy optimization*. In the *behavior policy disentangling* phase, we employ a CVAE to disentangle behavior policies by modeling the advantage values of collected state-action pairs as conditioned variables. The CVAE enables the agent to infer different action distributions associated with various behavior policies. Then in the *agent policy optimization* phase, the action distributions derived from the advantage condition serve as disentangled behavior policies, establishing an advantage-aware policy constraint to guide agent training. An overview of our A2PO is illustrated in Figure 2.

### 4.1 BEHAVIOR POLICY DISENTANGLING

To realize behavior policy disentangling, we adopt a CVAE to relate the distribution of the latent variable to that of the specific behavior policy under the given advantage-based condition variables. The CVAE model is consist of an encoder $q_\varphi(z|a,c)$ and a decoder $p_\psi(a|z,c)$, where $z$ denotes the latent variable and $c$ denotes the conditional variables. Concretely, the encoder $q_\varphi(z|a,c)$ is fed with condition $c$ and action $a$ to project them into a latent representation $z$. Given specific condition $c$ and the encoder output $z$, the decoder $p_\psi(a|z,c)$ captures the correlation between condition $c$ and latent representation $z$ to reconstruct the original action $a$. Unlike previous methods (Fujimoto et al., 2019; Chen et al., 2022; Wu et al., 2022) predicting action solely based on the state $s$, we consider both state $s$ and advantage value $\xi$ for CVAE condition. The state-advantage condition $c$ is formulated as:

$$c = s\,||\,\xi. \tag{3}$$

Therefore, given the current state $s$ and the advantage value $\xi$ as a joint condition, the CVAE model is able to generate corresponding action $a$ with varying quality positively correlated with the advantage condition $\xi$. For a state-action pair $(s,a)$, the advantage condition $\xi$ can be computed as follows:

$$\xi = \tanh\big(\min_{i=1,2} Q_{\theta_i}(s,a) - V_\phi(s)\big), \tag{4}$$

where two Q-networks with the $\min(\cdot)$ operation are adopted to ensure conservatism in offline RL settings (Fujimoto et al., 2019). Moreover, we employ the $\tanh(\cdot)$ function to normalize the advantage condition within the range of $(-1,1)$. This operation prevents excessive outliers from impacting the performance of CVAE, improving the controllability of generation. The optimization of the Q-networks and V-network will be described in the following section.

The CVAE model is trained using the state-advantage condition $c$ and the corresponding action $a$. The training objective involves maximizing the Empirical Lower Bound (ELBO) (Sohn et al., 2015)

on the log-likelihood of the sampled minibatch:

$$\mathcal{L}_{\text{CVAE}}(\varphi, \psi) = -\mathbb{E}_{\mathcal{D}} \left[ \mathbb{E}_{q_{\varphi}(z|a,c)} \left[ \log(p_{\psi}(a|z,c)) \right] + \alpha \cdot \text{KL} \left[ (q_{\varphi}(z|a,c) \parallel p(z)) \right] \right], \quad (5)$$

where $\alpha$ is the coefficient for trading off the KL-divergence loss term, and $p(z)$ denotes the prior distribution of $z$ setting to be $\mathcal{N}(0, 1)$. The first log-likelihood term encourages the generated action to match the real action as much as possible, while the second KL divergence term aligns the learned latent variable distribution with the prior distribution $p(z)$.

At each round of CVAE training, a minibatch of state-action pairs $(s, a)$ is sampled from the offline dataset. These pairs are fed to the critic network $Q_\theta$ and $V_\phi$ to get corresponding advantage condition $\xi$ by Equation (4). Then the advantage-aware CVAE is subsequently optimized by Equation (5). By incorporating the advantage condition $\xi$ into the CVAE, the CVAE captures the relation between $\xi$ and the action distribution of the behavior policy, as shown in the upper part of Figure 2. This enables the CVAE to generate actions $a$ based on the state-advantage condition $c$ in a manner where the action quality is positively correlated with $\xi$. Furthermore, the advantage-aware CVAE is utilized to establish an advantage-aware policy constraint for agent policy optimization in the next stage.

## 4.2 AGENT POLICY OPTIMIZATION

The agent is constructed using the actor-critic framework (Sutton & Barto, 2018). The critic comprises two Q-networks $Q_{\theta_{i=1,2}}$ and one V-network $V_\phi$. The advantage-aware policy $\pi_\omega(\cdot|c)$, with input $c = s \parallel \xi$, generates a latent representation $\tilde{z}$ based on the state $s$ and the designated advantage condition $\xi$. This latent representation $\tilde{z}$, along with $c$, is then fed into the decoder $p_\psi$ to decode a recognizable action $a_\xi$, as follows:

$$a_\xi \sim p_\psi(\cdot \mid \tilde{z}, c), \text{ where } \tilde{z} \sim \pi_\omega(\cdot \mid c). \quad (6)$$

The agent optimization, following the actor-critic framework, encompasses policy evaluation and policy improvement steps. During the policy evaluation step, the critic is optimized through the minimization of the temporal difference (TD) loss, as follows:

$$\mathcal{L}_{\text{Critic}}(\theta, \phi) = \mathbb{E}_{\substack{(s,a,r,s') \sim \mathcal{D} \\ \tilde{z}^* \sim \pi_\omega(\cdot|c^*), \\ a_\xi^* \sim p_\psi(\cdot|\tilde{z}^*, c^*)}} \left[ \sum_i \left[ r + V_{\hat{\phi}}(s) - Q_{\theta_i}(s,a) \right]^2 + \left[ r + \min_i Q_{\hat{\theta}_i}(s', a_\xi^*) - V_\phi(s) \right]^2 \right], \quad (7)$$

where $Q_{\hat{\theta}}$ and $V_{\hat{\phi}}$ are the target network updated softly, $a_\xi^*$ is obtained by the optimal policy $\pi_\omega(\cdot|c^*)$ and $c^* = s \parallel \xi^*$ is state $s$ enhanced with the largest advantage condition $\xi^* = 1$ since the range of $\xi$ is normalized in Equation (4). The first term of $\mathcal{L}_{\text{Critic}}$ is to optimize Q-network while the second term is to optimize V-network. Different from Equation (1), we introduce the target Q and V networks to directly optimize the mutual online network to stabilize the critic training.

For the policy improvement step, the TD3BC-style (Fujimoto & Gu, 2021) loss is defined as:

$$\mathcal{L}_{\text{Actor}}(\omega) = -\lambda \mathbb{E}_{\substack{s \sim \mathcal{D}, \\ \tilde{z}^* \sim \pi_\omega(\cdot|c^*), \\ a_\xi^* \sim p_\psi(\cdot|\tilde{z}^*, c^*)}} Q_{\theta_1}(s, a_\xi^*) + \mathbb{E}_{\substack{(s,a) \sim \mathcal{D}, \\ \tilde{z} \sim \pi_\omega(\cdot|c), \\ a_\xi \sim p_\psi(\cdot|\tilde{z},c)}} (a - a_\xi)^2, \quad (8)$$

where $a_\xi^*$ is the optimal action sampled from $p_\psi(\cdot|\tilde{z}^*, c^*)$ with $\pi_\omega(\cdot|c^*)$ and $c^* = s \parallel \xi^*$, and advantage condition $\xi$ for $a_\xi$ in the second term is obtained from the critic by Equation (4). Meanwhile, following TD3BC (Fujimoto & Gu, 2021), we add a normalization coefficient $\lambda = \alpha/(\frac{1}{N} \sum_{(s_i, a_i)} |Q(s_i, a_i)|)$ to the first term to keep the scale balance between Q value objective and regularization, where $\alpha$ is a hyperparameter to control the scale of the normalized Q value. The first term encourages the optimal policy condition on $c^*$ to select actions that yield the highest expected returns represented by the Q-value. This aligns with the policy improvement step commonly seen in conventional reinforcement learning approaches. The second behavior cloning term explicitly imposes constraints on the advantage-aware policy, ensuring the policy selects in-sample actions that adhere to the advantage condition $\xi$ determined by the critic. Therefore, the suboptimal samples with low advantage condition $\xi$ will not disrupt the optimization of optimal policy $\pi_\omega(\cdot|c^*)$. And they enforce valid constraints on the corresponding policy $\pi_\omega(\cdot|c)$, as shown in the lower part of Figure 2. It should be noted that the decoder $p_\psi$ is fixed during both policy evaluation and improvement.

To make A2PO clearer for readers, the pseudocode is provided in Appendix A. It is important to note that while the CVAE and the agent are trained in an alternating manner, the CVAE training step $K$ is much less than the total training step $T$. This disparity arises from the fact that, as the training progresses, the critic $Q_\theta$ and $V_\phi$ gradually converge towards their optimal values. Consequently, the computed advantage conditions $\xi$ of most transitions tend to be negative, except a small portion of superior ones with positive $\xi$. And the low values of $\xi$ are insufficient to enforce policy optimization. Therefore, as training progresses further, it becomes essential to keep the advantage-aware CVAE fixed to ensure stable policy optimization, and we will illustrate this conclusion in Section 5.

## 5 EXPERIMENTS

To illustrate the effectiveness of the proposed A2PO method, we conduct experiments on the D4RL benchmark (Fu et al., 2020). We aim to answer the following questions: (1) Can A2PO outperform the state-of-the-art offline RL methods in both the single-quality datasets and mixed-quality datasets? (Section 5.2 and Appendix C, I, J) (2) How do different components of A2PO contribute to the overall performance? (Section 5.3 and Appendix D–G, K) (3) Can the A2PO agent effectively estimate the advantage value of different transitions? (Section 5.4 and Appendix H) (4) How does A2PO perform under mixed-quality datasets with varying single-quality samples? (Appendix L)

### 5.1 EXPERIMENT SETTINGS

**Tasks and Datasets.** We evaluate the proposed A2PO on three locomotion tasks (*i.e., halfcheetah-v2, walker2d-v2,* and *hopper-v2*) and six navigation tasks (*i.e., maze2d-umaze-v1, maze2d-medium-v1, maze2d-large-v1, antmaze-umaze-diverse-v1, antmaze-medium-diverse-v1,* and *antmaze-large-diverse-v1*) using the D4RL benchmark (Fu et al., 2020). For each locomotion task, we conduct experiments using both the single-quality and mixed-quality datasets. The single-quality datasets are generated with the *random, medium,* and *expert* behavior policies. The mixed-quality datasets are combinations of these single-quality datasets, including *medium-expert, medium-replay, random-medium, medium-expert,* and *random-medium-expert.* Since the D4RL benchmark only includes the first two mixed-quality datasets, we manually construct the last three mixed-quality datasets by directly combining the corresponding single-quality datasets. For each navigation task, we directly adopt the single-quality dataset in the D4RL benchmark generated by the *expert* behavior policy.

**Comparison Methods and Hyperparameters.** We compare the proposed A2PO to several state-of-the-art offline RL methods: BCQ (Fujimoto et al., 2019), TD3BC (Fujimoto & Gu, 2021), CQL (Kumar et al., 2020), IQL (Kostrikov et al., 2021), especially the advantage-weighted offline RL methods: LAPO (Chen et al., 2022) and BPPO (Zhuang et al., 2023). Besides, we also select the vanilla BC method (Pomerleau, 1991) and the model-based offline RL method, MOPO (Yu et al., 2020), for comparison. The detailed hyperparameters are given in Appendix B.2.

### 5.2 COMPARISON ON D4RL BENCHMARKS

**Locomotion.** The experimental results of all compared methods in D4RL locomotion tasks are presented in Table 1. With the single-quality dataset in the locomotion tasks, our A2PO achieves state-of-the-art results with low variance across most tasks. Moreover, both AW method LAPO and non-AW baselines like TD3BC and even BC achieve acceptable performance, which indicates that the conflict issue hardly occurs in the single-quality dataset. As for the D4RL mixed-quality dataset *medium-expert* and *medium-replay*, the performance of other baselines shows varying degrees of degradation. Particularly, the non-AW methods are particularly affected, as seen in the performance gap of BC and BCQ between *hopper-expert* and *hopper-medium-expert*. AW method LAPO remains relatively excellent, while our A2PO continues to achieve the best performance on these datasets. Notably, some baselines, such as MOPO, show improved results due to the gain of samples from behavior policies, leading to a more accurate reconstruction of the environmental dynamics. The newly constructed mixed-quality datasets, namely *random-medium, random-expert,* and *random-medium-expert*, highlight the issue of substantial gaps between behavior policies. The results reveal a significant drop in performance and increased variance for all other baselines, including the AW methods LAPO and BPPO. However, our A2PO consistently outperforms most other baselines on the majority of these datasets. When considering the total scores across all datasets, A2PO outper-

Table 1: Test returns of our proposed A2PO and baselines on the locomotion tasks. $\pm$ corresponds to one standard deviation of the average evaluation of the performance on 5 random seeds. The performance is measured by the normalized scores at the last training iteration. **Bold** indicates the best performance in each task. Corresponding learning curves are reported in Appendix C.

| Source | Task | BC | BCQ | TD3BC | CQL | IQL | MOPO | BPPO | LAPO | A2PO (Ours) |
|---|---|---|---|---|---|---|---|---|---|---|
| random | halfcheetah | $2.25_{\pm0.00}$ | $2.25_{\pm0.00}$ | $11.25_{\pm0.97}$ | $23.41_{\pm0.73}$ | $14.52_{\pm2.87}$ | $\mathbf{37.75}_{\pm3.45}$ | $2.25_{\pm0.00}$ | $26.99_{\pm0.78}$ | $25.52_{\pm0.98}$ |
| | hopper | $3.31_{\pm0.85}$ | $7.43_{\pm0.48}$ | $9.2_{\pm1.75}$ | $2.97_{\pm2.49}$ | $8.18_{\pm0.59}$ | $18.19_{\pm9.65}$ | $2.56_{\pm0.07}$ | $16.08_{\pm7.30}$ | $\mathbf{18.43}_{\pm0.42}$ |
| | walker2d | $1.63_{\pm0.54}$ | $4.43_{\pm0.95}$ | $1.07_{\pm1.04}$ | $2.09_{\pm1.85}$ | $6.47_{\pm1.16}$ | $0.04_{\pm0.02}$ | $\mathbf{6.53}_{\pm0.64}$ | $1.91_{\pm1.32}$ | $3.59_{\pm1.74}$ |
| medium | halfcheetah | $42.14_{\pm0.33}$ | $46.83_{\pm0.18}$ | $\mathbf{48.31}_{\pm0.10}$ | $47.20_{\pm0.20}$ | $47.63_{\pm0.05}$ | $48.40_{\pm0.17}$ | $0.62_{\pm0.81}$ | $45.58_{\pm0.06}$ | $47.09_{\pm0.17}$ |
| | hopper | $50.45_{\pm2.31}$ | $56.37_{\pm2.74}$ | $58.55_{\pm1.17}$ | $74.20_{\pm0.82}$ | $51.17_{\pm2.62}$ | $5.68_{\pm4.00}$ | $13.42_{\pm9.17}$ | $52.53_{\pm2.61}$ | $\mathbf{80.29}_{\pm3.95}$ |
| | walker2d | $71.73_{\pm2.44}$ | $73.12_{\pm1.38}$ | $83.62_{\pm0.85}$ | $80.38_{\pm0.77}$ | $63.75_{\pm3.91}$ | $0.09_{\pm0.06}$ | $31.38_{\pm18.57}$ | $80.46_{\pm1.25}$ | $\mathbf{84.88}_{\pm0.23}$ |
| expert | halfcheetah | $92.02_{\pm0.32}$ | $92.69_{\pm0.94}$ | $96.74_{\pm0.37}$ | $5.58_{\pm2.96}$ | $92.37_{\pm1.45}$ | $9.68_{\pm2.43}$ | $5.51_{\pm2.02}$ | $95.33_{\pm0.17}$ | $\mathbf{96.26}_{\pm0.27}$ |
| | hopper | $104.56_{\pm3.51}$ | $77.58_{\pm4.14}$ | $108.61_{\pm1.47}$ | $93.95_{\pm1.98}$ | $69.81_{\pm12.86}$ | $5.37_{\pm4.09}$ | $2.56_{\pm2.05}$ | $110.45_{\pm0.89}$ | $\mathbf{111.70}_{\pm0.39}$ |
| | walker2d | $108.61_{\pm0.19}$ | $110.13_{\pm0.28}$ | $110.13_{\pm0.05}$ | $105.62_{\pm0.94}$ | $108.53_{\pm0.76}$ | $23.21_{\pm13.04}$ | $4.54_{\pm0.40}$ | $111.55_{\pm0.11}$ | $\mathbf{112.36}_{\pm0.23}$ |
| medium replay | halfcheetah | $18.97_{\pm13.85}$ | $40.87_{\pm0.21}$ | $44.51_{\pm0.22}$ | $\mathbf{46.74}_{\pm0.13}$ | $43.99_{\pm0.33}$ | $37.46_{\pm28.06}$ | $11.82_{\pm7.17}$ | $41.94_{\pm0.47}$ | $44.74_{\pm0.22}$ |
| | hopper | $20.99_{\pm3.92}$ | $48.19_{\pm5.52}$ | $65.20_{\pm9.77}$ | $91.34_{\pm1.99}$ | $52.61_{\pm3.61}$ | $75.05_{\pm28.82}$ | $12.68_{\pm6.57}$ | $50.14_{\pm11.16}$ | $\mathbf{101.59}_{\pm1.25}$ |
| | walker2d | $13.99_{\pm6.71}$ | $52.62_{\pm4.62}$ | $81.28_{\pm3.12}$ | $79.93_{\pm1.26}$ | $68.84_{\pm8.39}$ | $60.68_{\pm19.32}$ | $3.17_{\pm3.05}$ | $60.55_{\pm10.45}$ | $\mathbf{82.82}_{\pm1.70}$ |
| medium expert | halfcheetah | $45.18_{\pm1.22}$ | $46.87_{\pm0.18}$ | $91.52_{\pm1.82}$ | $16.47_{\pm3.62}$ | $87.71_{\pm1.97}$ | $69.73_{\pm6.67}$ | $21.02_{\pm14.44}$ | $94.22_{\pm0.46}$ | $\mathbf{95.61}_{\pm0.54}$ |
| | hopper | $54.44_{\pm4.05}$ | $58.05_{\pm4.03}$ | $98.58_{\pm2.48}$ | $89.19_{\pm12.15}$ | $36.04_{\pm21.36}$ | $20.32_{\pm13.22}$ | $16.28_{\pm2.66}$ | $111.04_{\pm0.36}$ | $\mathbf{107.44}_{\pm0.56}$ |
| | walker2d | $90.54_{\pm5.93}$ | $75.14_{\pm1.18}$ | $110.28_{\pm0.26}$ | $102.65_{\pm3.13}$ | $104.13_{\pm0.76}$ | $91.92_{\pm7.63}$ | $13.28_{\pm12.31}$ | $110.88_{\pm0.15}$ | $\mathbf{112.13}_{\pm0.24}$ |
| random medium | halfcheetah | $2.25_{\pm0.00}$ | $12.71_{\pm3.89}$ | $47.71_{\pm0.07}$ | $31.89_{\pm16.67}$ | $42.23_{\pm0.95}$ | $\mathbf{52.71}_{\pm4.27}$ | $2.25_{\pm0.00}$ | $18.53_{\pm0.99}$ | $45.20_{\pm0.21}$ |
| | hopper | $\mathbf{23.20}_{\pm8.00}$ | $9.24_{\pm0.77}$ | $7.42_{\pm3.17}$ | $3.33_{\pm3.59}$ | $6.18_{\pm0.66}$ | $19.86_{\pm12.21}$ | $9.14_{\pm11.23}$ | $4.17_{\pm3.11}$ | $7.14_{\pm0.35}$ |
| | walker2d | $19.16_{\pm18.96}$ | $0.20_{\pm0.27}$ | $10.68_{\pm0.57}$ | $0.19_{\pm0.63}$ | $54.58_{\pm2.21}$ | $40.18_{\pm33.10}$ | $21.96_{\pm22.91}$ | $23.65_{\pm33.97}$ | $\mathbf{75.80}_{\pm2.12}$ |
| random expert | halfcheetah | $13.73_{\pm18.94}$ | $2.10_{\pm1.48}$ | $43.05_{\pm8.57}$ | $15.03_{\pm11.68}$ | $28.64_{\pm7.90}$ | $18.50_{\pm2.31}$ | $2.24_{\pm0.00}$ | $52.58_{\pm17.30}$ | $\mathbf{90.32}_{\pm1.63}$ |
| | hopper | $10.14_{\pm10.75}$ | $8.53_{\pm3.62}$ | $78.81_{\pm25.50}$ | $7.75_{\pm6.91}$ | $58.50_{\pm12.86}$ | $17.15_{\pm3.80}$ | $11.22_{\pm11.98}$ | $82.33_{\pm18.95}$ | $\mathbf{105.19}_{\pm4.54}$ |
| | walker2d | $14.70_{\pm11.35}$ | $0.56_{\pm0.89}$ | $6.96_{\pm1.73}$ | $0.27_{\pm0.78}$ | $90.88_{\pm9.99}$ | $4.56_{\pm6.06}$ | $1.47_{\pm2.26}$ | $0.39_{\pm0.53}$ | $\mathbf{91.96}_{\pm10.98}$ |
| random medium expert | halfcheetah | $2.25_{\pm0.01}$ | $15.91_{\pm7.32}$ | $62.33_{\pm4.96}$ | $13.50_{\pm12.12}$ | $61.61_{\pm4.11}$ | $26.72_{\pm8.34}$ | $2.19_{\pm0.02}$ | $71.09_{\pm0.47}$ | $\mathbf{90.58}_{\pm1.44}$ |
| | hopper | $27.35_{\pm5.79}$ | $3.99_{\pm3.55}$ | $60.51_{\pm35.16}$ | $9.43_{\pm6.36}$ | $57.88_{\pm13.77}$ | $13.30_{\pm8.45}$ | $16.00_{\pm8.21}$ | $66.59_{\pm19.29}$ | $\mathbf{107.84}_{\pm0.42}$ |
| | walker2d | $24.57_{\pm9.34}$ | $2.39_{\pm2.46}$ | $15.71_{\pm3.87}$ | $0.05_{\pm0.21}$ | $90.83_{\pm5.10}$ | $56.39_{\pm19.57}$ | $21.26_{\pm9.54}$ | $60.41_{\pm43.32}$ | $\mathbf{97.71}_{\pm6.74}$ |
| Total | | $768.43_{\pm38.36}$ | $848.20_{\pm14.22}$ | $1352.03_{\pm46.23}$ | $943.16_{\pm29.40}$ | $1347.08_{\pm36.20}$ | $752.94_{\pm66.53}$ | $235.35_{\pm43.21}$ | $1389.39_{\pm66.12}$ | $\mathbf{1837.19}_{\pm14.88}$ |

Table 2: Test returns of our proposed A2PO and baselines on the navigation tasks.

| Task | BC | BCQ | TD3BC | CQL | IQL | MOPO | BPPO | LAPO | A2PO (Ours) |
|---|---|---|---|---|---|---|---|---|---|
| maze2d-u | $0.46_{\pm2.92}$ | $24.79_{\pm1.15}$ | $24.19_{\pm20.80}$ | $17.02_{\pm1.87}$ | $56.17_{\pm9.86}$ | $-15.40_{\pm0.53}$ | $14.02_{\pm1.03}$ | $78.00_{\pm9.93}$ | $\mathbf{133.27}_{\pm9.58}$ |
| maze2d-m | $0.73_{\pm1.35}$ | $22.51_{\pm11.38}$ | $33.50_{\pm23.70}$ | $22.45_{\pm6.70}$ | $25.67_{\pm16.93}$ | $19.09_{\pm14.23}$ | $3.22_{\pm1.50}$ | $43.21_{\pm0.85}$ | $\mathbf{83.95}_{\pm10.56}$ |
| maze2d-l | $1.11_{\pm1.06}$ | $42.95_{\pm10.17}$ | $\mathbf{128.46}_{\pm29.62}$ | $2.53_{\pm6.58}$ | $45.67_{\pm18.91}$ | $-0.53_{\pm1.40}$ | $2.45_{\pm5.68}$ | $69.70_{\pm2.39}$ | $127.61_{\pm5.35}$ |
| antmaze-u-d | $50.00_{\pm2.83}$ | $53.33_{\pm12.47}$ | $60.00_{\pm43.20}$ | $80.00_{\pm8.16}$ | $86.67_{\pm12.47}$ | $0.00_{\pm0.00}$ | $24.00_{\pm14.24}$ | $84.13_{\pm4.11}$ | $\mathbf{96.66}_{\pm4.71}$ |
| antmaze-m-d | $0.00_{\pm0.00}$ | $6.67_{\pm4.71}$ | $3.33_{\pm4.71}$ | $0.00_{\pm0.00}$ | $46.67_{\pm18.86}$ | $0.00_{\pm0.00}$ | $0.00_{\pm0.00}$ | $1.18_{\pm0.94}$ | $\mathbf{50.00}_{\pm15.25}$ |
| antmaze-l-d | $0.00_{\pm0.00}$ | $0.00_{\pm0.00}$ | $0.00_{\pm0.00}$ | $6.67_{\pm4.71}$ | $\mathbf{43.33}_{\pm12.47}$ | $0.00_{\pm0.00}$ | $0.00_{\pm0.00}$ | $0.00_{\pm0.00}$ | $6.00_{\pm4.90}$ |
| Total | $52.30_{\pm7.15}$ | $144.25_{\pm18.69}$ | $249.48_{\pm60.30}$ | $128.67_{\pm13.43}$ | $304.18_{\pm37.53}$ | $3.16_{\pm14.31}$ | $43.69_{\pm17.31}$ | $276.22_{\pm12.78}$ | $\mathbf{497.49}_{\pm22.60}$ |

forms the next best-performing and recently state-of-the-art AW method, LAPO, by over 33%. This comparison highlights the superior performance achieved by A2PO, showcasing its effective disentanglement of action distributions from different behavior policies in order to enforce a reasonable advantage-aware policy constraint and obtain an optimal agent policy.

**Navigation.** Table 2 presents the experimental results of all the compared methods on the D4RL navigation tasks. Among the offline RL baselines and AW methods, A2PO demonstrates remarkable performance in the challenging maze navigation tasks, showcasing the robust representation capabilities of the advantage-aware policy.

## 5.3 ABLATION ANALYSIS

**Different Advantage condition during training.** The performance comparison of different advantage condition computing methods for agent training is given in Figure 3. Equation (4) obtains continuous advantage condition $\xi$ in the range of $(-1, 1)$. To evaluate the effectiveness of the continuous computing method, we design a discrete form of advantage condition: $\xi_{\text{dis}} = \text{sgn}(\xi) \cdot \mathbf{1}_{|\xi|>\epsilon}$, where $\text{sgn}(\cdot)$ is the symbolic function, and $\mathbf{1}_{|\xi|>\epsilon}$ is the indicator function returning 1 if the absolute value of $\xi$ is greater than the hyperparameter of threshold $\epsilon$, otherwise 0. Thus, the advantage condition $\xi_{\text{dis}}$ is constrained to discrete value of $\{-1, 0, 1\}$. Moreover, if threshold $\epsilon = 0$, $\xi_{\text{dis}}$ only takes values from $\{-1, 1\}$. Another special form of advantage condition is $\xi_{\text{fix}} = 1$ for all state-action pairs, in which the advantage-aware ability is lost. Figure 3a shows that setting $\xi_{\text{fix}} = 1$ without

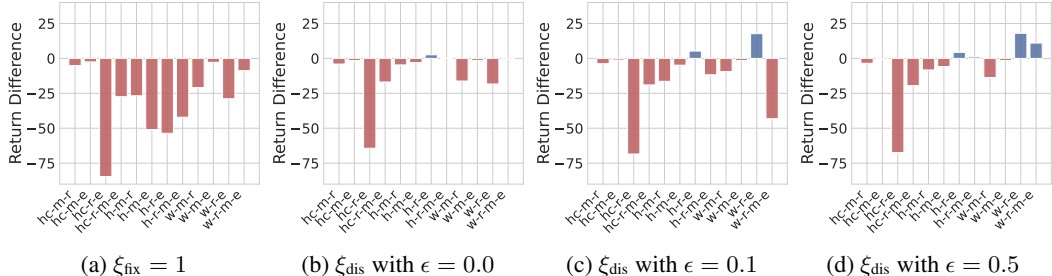

(a) $\xi_{\text{fix}} = 1$  (b) $\xi_{\text{dis}}$ with $\epsilon = 0.0$  (c) $\xi_{\text{dis}}$ with $\epsilon = 0.1$  (d) $\xi_{\text{dis}}$ with $\epsilon = 0.5$

Figure 3: Test return difference of A2PO with different discrete advantage conditions during training compared with original A2PO with continuous advantage condition during training. The full forms of task abbreviations are listed in Appendix B.1. Detailed test returns are reported in Appendix D.

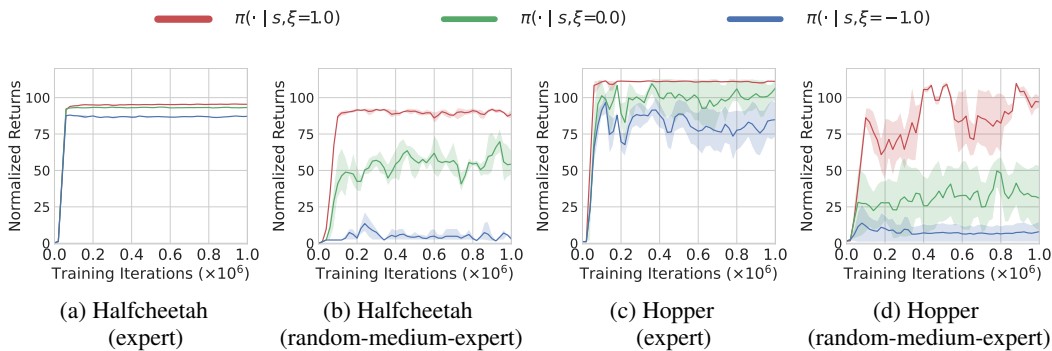

(a) Halfcheetah (expert)  (b) Halfcheetah (random-medium-expert)  (c) Hopper (expert)  (d) Hopper (random-medium-expert)

Figure 4: Learning curves of A2PO with different discrete advantage conditions for test while using the original continuous advantage condition during training. Test returns are reported in Appendix E.

explicitly advantage-aware mechanism leads to a significant performance decreasing, especially in the new mixed-quality dataset. Meanwhile, $\xi_{\text{dis}}$ with different values of threshold $\epsilon$ achieve slightly inferior results than the continuous $\xi$. This outcome strongly supports the efficiency of behavior policy disentangling. Although $\xi_{\text{dis}}$ input makes this process easier, $\xi_{\text{dis}}$ hiddens the concrete advantage value, causing a mismatch between the advantage value and the sampled transition.

**Different Advantage condition for test.** The performance comparison of different discrete advantage conditions for test is given in Figure 4. To ensure clear differentiation, we select the advantage conditions $\xi$ from $\{-1, 0, 1\}$. The different designated advantage conditions $\xi$ are fixed input for the actor, leading to different policies $\pi_\omega(\cdot|s, \xi)$. The final outcomes demonstrate the partition of returns corresponding to the policies with different $\xi$. Furthermore, the magnitude of the gap increases as the offline dataset includes samples from more behavior policies. These observations provide strong evidence for the success of A2PO disentangling the behavior policies under the multi-quality dataset.

**Different CVAE training steps.** The results of different CVAE training step $K$ is presented in Figure 5. The results show that $K = 2 \times 10^5$ achieves the overall best average performance, while both $K = 10^5$ and $K = 10^6$ exhibit higher variances or larger fluctuations. For $K = 10^5$, A2PO converges to a quite good level but not as excellent as $K = 2 \times 10^5$. In this case, the behavior policies disentangling halt prematurely, leading to incomplete CVAE learning. For $K = 10^6$, high returns are typically achieved at the early stages but diminish significantly later. This can be attributed to the fact that as the critic being optimized, the critic assigns high advantage conditions to only a small portion of transitions. The resulting unbalanced distribution of advantage conditions hampers the learning of both the advantage-aware CVAE and policy.

## 5.4 VISUALIZATION

Figure 6 presents the visualization of A2PO latent representation. The uniformly sampled advantage condition $\xi$ combined with the initial state $s$, are fed into the actor network to get the latent repre-

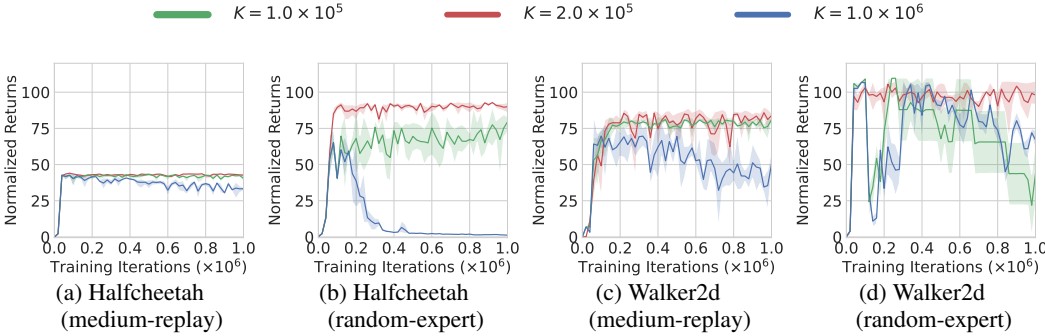

Figure 5: Learning curves of A2PO with different CVAE training steps (*i.e.*, the number of training iterations for CVAE optimization). Test returns are reported in Appendix F.

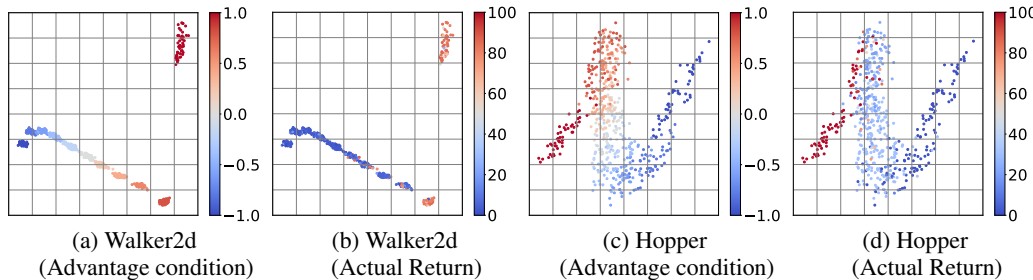

Figure 6: Visualization of A2PO latent representation after applying PCA with different advantage conditions and actual returns in the *walker2d-medium-replay* and *hopper-medium-replay* tasks. Each data point indicates a latent representation $\tilde{z}$ based on the initial state and different advantage conditions sampled uniformly from $[-1, 1]$. The actual return is measured under the corresponding sampled advantage condition. The value magnitude is indicated with varying shades of color.

sentation generated by the final layer of the actor. The result demonstrates that the representations converge according to the advantage and the actual return. Notably, the return of each point aligns with the corresponding variations in $\xi$. Moreover, as $\xi$ increases monotonically, the representations undergo continuous alterations in a rough direction. These observations suggest the effectiveness of advantage-aware policy construction. Meanwhile, more experiments of advantage estimation conducted on different tasks and datasets are presented in Appendix H.

# 6 CONCLUSION

In this paper, we propose a novel approach, termed as A2PO, to tackle the constraint conflict issue on mixed-quality offline dataset with advantage-aware policy constraint. Specifically, A2PO utilizes a CVAE to effectively disentangle the action distributions associated with various behavior policies. This is achieved by modeling the advantage values of all training data as conditional variables. Consequently, advantage-aware agent policy optimization can be focused on maximizing high advantage values while conforming to the disentangled distribution constraint imposed by the mixed-quality dataset. Experimental results show that A2PO successfully decouples the underlying behavior policies and significantly outperforms state-of-the-art offline RL competitors. One limitation of A2PO is the instability of the advantage condition computed by the critic networks. As the training progresses, the critic is optimized continuously, measuring the same transition with distinct advantage conditions. The instability of the advantage condition poses challenges for both CVAE and agent training. To address this issue, we halt the CVAE training after a predetermined number of training steps to prevent performance collapse, which heavily relies on the specified step number. To overcome this limitation, our future work will focus on extending A2PO to design an adaptive advantage condition computing mechanism for stable training.

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

# A  METHOD

To make the proposed Advantage-Aware Policy Optimization (A2PO) method clearer for readers, the pseudocode is provided in Algorithm 1.

---

**Algorithm 1** Advantage-Aware Policy Optimization (A2PO)

---

**Input:** offline dataset $\mathcal{D}$, CVAE training step $K$, total training step $T$, soft update rate $\tau$.
**Initialize:** CVAE encoder $q_\varphi$ and decoder $p_\psi$, actor network $\pi_\omega$, critic networks $Q_\theta$ and $V_\phi$.

   **for** $i = 1$ **to** $T$ **do**
      Sample random minibatch of transitions $\mathcal{B} = \{(s, a, r, s')\} \sim \mathcal{D}$.
      $\xi = \tanh(\min_{i=1,2} Q_{\theta_i}(s, a) - V_\phi(s)), \xi^* = 1, c = s||\xi, c^* = s||\xi^*$
      # Behavior Policy Disentangling
      **if** $i \leq K$ **then**
         Optimize CVAE encoder $q_\varphi$ and decoder $p_\psi$ by

$$\mathcal{L}_{\text{CVAE}}(\varphi, \psi) = -\mathbb{E}_\mathcal{D} \left[ \mathbb{E}_{q_\varphi(z|a,c)} \left[ \log(p_\psi(a|z,c)) \right] + \alpha \cdot \text{KL} \left[ (q_\varphi(z|a,c) \parallel p(z)) \right] \right].$$

      **end if**
      # Agent Policy Optimization
      Optimize critic networks $Q_\theta$ and $V_\phi$ by

$$\mathcal{L}_{\text{Critic}}(\theta, \phi) = \mathbb{E}_{\substack{(s,a,r,s')\sim\mathcal{D} \\ \tilde{z}^*\sim\pi_\omega(\cdot|c^*), \\ a_\xi^*\sim p_\psi(\cdot|\tilde{z}^*,c^*)}} \left[ \sum_i \left[ r + V_{\hat{\phi}}(s) - Q_{\theta_i}(s, a) \right]^2 + \left[ r + \min_i Q_{\hat{\theta}_i}(s', a_\xi^*) - V_\phi(s) \right]^2 \right].$$

      Optimize actor network $\pi_\omega$ by

$$\mathcal{L}_{\text{Actor}}(\omega) = -\lambda\mathbb{E}_{\substack{s\sim\mathcal{D}, \\ \tilde{z}^*\sim\pi_\omega(\cdot|c^*), \\ a_\xi^*\sim p_\psi(\cdot|\tilde{z}^*,c^*)}} Q_{\theta_1}(s, a_\xi^*) + \mathbb{E}_{\substack{(s,a)\sim\mathcal{D}, \\ \tilde{z}\sim\pi_\omega(\cdot|c), \\ a_\xi\sim p_\psi(\cdot|\tilde{z},c)}} (a - a_\xi)^2.$$

      Soft-update the target network: $\hat{\theta} \leftarrow (1 - \tau)\hat{\theta} + \tau\theta, \hat{\phi} \leftarrow (1 - \tau)\hat{\phi} + \tau\phi$.
   **end for**

---

# B  EXPERIMENT DETAILS

## B.1  TASK ABBREVIATION

In order to improve the readability and conciseness, we adopt abbreviations for the tasks throughout the main text. The corresponding abbreviations for each task are provided in Table S1 and Table S2.

## B.2  IMPLEMENTATION DETAILS

In this section, we provide the implementation details of our experiments. We conducted our experiments using PyTorch 3.8 (Paszke et al., 2019) on a cluster of 8 A6000 GPUs. Each run required approximately 8 hours to complete 1 million steps. The source code will be made openly available upon the publication of this paper. For our experiments, we utilized fixed and selectable hyperparameters, presented in Table S3 and Table S4 respectively. Following the TD3BC approach, we incorporated Q normalization, policy noise, and policy clipping during the training process. For the hyperparameter value $\alpha$ in Q normalization, we follow the same setting $\alpha = 2.5$ as in the official TD3BC implementation and keep this hyperparameter on the whole experiment. These techniques have been demonstrated to significantly enhance performance (Fujimoto & Gu, 2021). The CVAE was optimized over $K$ step, while the actor-critic model was trained for $T$ step. The critic network

Table S1: The abbreviation of the corresponding locomotion task and dataset.

| Dataset | Halfcheetah-v2 | Hopper-v2 | Walker2d-v2 |
|---|---|---|---|
| random | hc-r | h-r | w-r |
| medium | hc-m | h-m | w-m |
| expert | hc-e | h-e | w-e |
| medium-replay | hc-m-r | h-m-r | w-m-r |
| medium-expert | hc-m-e | h-m-e | w-m-e |
| random-medium | hc-r-m | h-r-m | w-r-m |
| random-expert | hc-m-e | h-m-e | w-m-e |
| random-medium-expert | hc-r-m-e | h-r-m-e | w-r-m-e |

Table S2: The abbreviation of the corresponding navigation task.

| Task & Dataset | Abbreviation |
|---|---|
| maze2d-umaze-v1 | maze2d-u |
| maze2d-medium-v1 | maze2d-m |
| maze2d-large-v1 | maze2d-l |
| antmaze-umaze-diverse-v1 | antmaze-u-d |
| antmaze-medium-diverse-v1 | antmaze-m-d |
| antmaze-large-diverse-v1 | antmaze-l-d |

was updated at each step, whereas the actor network and the target critic networks were updated once after specific steps of critic optimization. By employing the default hyperparameters in Table S3 and the specific hyperparameters outlined in Table S4, the A2PO method achieved state-of-the-art performance across multiple tasks, as mentioned in Section 5.

As for the implementation of other baselines, the BC baseline is implemented based on the BPPO implementation available at: github.com/Dragon-Zhuang/BPPO. The CQL, IQL and MOPO baselines are implemented using the implementations provided at github.com/young-geng/cql, gwthomas/iql-pytorch, and github.com/yihaosun1124/OfflineRL-Kit, respectively. The remaining baselines, including BCQ, TD3BC, MOPO, BPPO, and LAPO, are implemented using the original implementations provided by the authors of the respective papers. These implementations can be found at: BCQ github.com/sfujim/BCQ, TD3BC github.com/sfujim/TD3_BC, BPPO github.com/Dragon-Zhuang/BPPO, and LAPO github.com/pcchenxi/LAPO-offlienRL.

## C  BASELINES COMPARISON

We plot the learning curves of locomotion tasks in Figure S1 and Figure S2, while the curves of navigation tasks are in Figure S3. The performance is evaluated every 20000 steps for each random seed. This assessment is based on the execution of 10 complete trajectories using the current policy. Compared with the state-of-the-art baseline methods, our proposed A2PO significantly improves the final performance. The results demonstrate that the A2PO agent has successfully obtained the optimal policy across diverse behavior policies.

## D  ANALYSIS OF ADVANTAGE CONDITION INPUT FOR TRAINING

Learning curves of our proposed A2PO method and baselines on the navigation tasks under the single-quality *expert* dataset. In this section, we provide a full comparison of different advantage condition computing methods for training on Locomotion tasks in Table S5. These computing methods are thoroughly described in Section 5.3. From detailed data, all of the discrete advantage conditions suffer from unstable performance as well as large variance. In *hopper-random-medium-expert* and *walker2d-random-medium-expert*, $\xi_{\text{dis}}$ with $\epsilon = 0.5$ works the best. And in *hopper-random-expert* and *walker2d-random-expert*, $\xi_{\text{dis}}$ with $\epsilon = 0.1$ works the best. However, in most cases, continuous advantage condition $\xi$ works well. Thus we use continuous $\xi$ by default for simplicity.

Table S3: The default hyperparameters in A2PO.

| | Hyperparameters | Value |
|---|---|---|
| CVAE and actor-critic hyperparameter | Total training step $T$ | $1 * 10^6$ |
| | CVAE training step $K$ | $2 * 10^5$ |
| | Soft update rate $\tau$ | 0.005 |
| | Whether use discrete $\xi$ | False |
| | Batch size | 256 |
| | Policy noise | 0.2 |
| | Policy clip range | $[-0.5, 0.5]$ |
| | Q normalization | 2.5 |
| | State normalization | True |
| | Actor update frequency | 2 |
| | Optimizer | Adam |
| | CVAE learning rate | $3 * 10^{-4}$ |
| | Actor learning rate | $3 * 10^{-4}$ |
| | Critic learning rate | $3 * 10^{-4}$ |
| | CVAE loss coefficient | 0.5 |
| Network architecture | Actor hidden layer | $[256, 256]$ |
| | Critic hidden layer | $[256, 256, 256]$ |
| | CVAE encoder hidden layer | $[750, 750]$ |
| | CVAE decoder hidden layer | $[750, 750]$ |
| | Latent space dimension | $2 * |\mathcal{A}|$ |

Table S4: The specific hyperparameters in A2PO.

| Hyperparameter | Task | Value |
|---|---|---|
| CVAE training step $K$ | Antmaze-large-diverse-v1 | $1 * 10^5$ |
| | Maze2d-umaze-v1 | $4 * 10^5$ |
| | maze2d-medium-v1 | $4 * 10^5$ |
| | Others | $2 * 10^5$ |
| Whether use discrete $\xi$ | Maze2d-medium-v1 | True, $\epsilon = 0.1$ |
| | Antmaze-large-diverse-v1 | True, $\epsilon = 0.3$ |
| | Others | False |

## E  ANALYSIS OF ADVANTAGE CONDITION INPUT FOR TESTING

In this section, we provide a full comparison of different fixed advantage condition inputs for testing on locomotion tasks in Table S6 as a supplement for Figure 4. The result shows that the agent performance improves as the input fixed advantage condition $\xi$ increases.

## F  ANALYSIS OF CVAE TRAIN STEPS

In this section, we consider exploring the influence of the CVAE train step $K$. We provide a full comparison of different CVAE train steps on locomotion tasks in Table S7. From Table S7, we can observe that the performance with large train step $K = 1 * 10^6$ or with low train step $K = 1 * 10^5$ is unstable over these tasks and even crash after training in the *halfcheetah-random-expert* and *walker2d-random-expert* tasks. Thus, we select the moderate $K = 2 * 10^5$ to achieve a stable mixed-quality behavior policy capture by default.

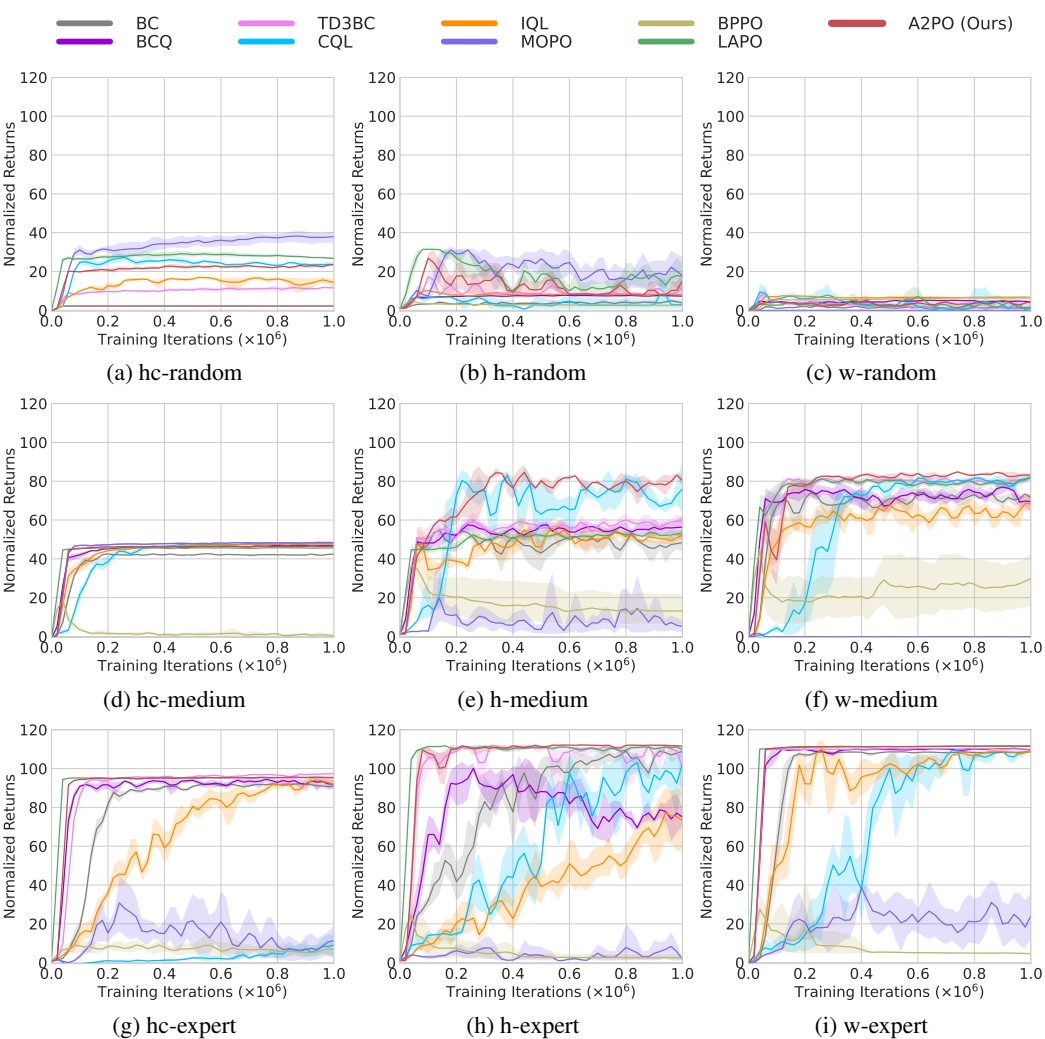

Figure S1: Learning curves of our proposed A2PO and baselines on the locomotion tasks under different single-quality datasets. "hc" denotes *halfcheetah*, "h" denotes *hopper*, and "w" denotes *walker2d*. All experimental results are illustrated with the mean and the standard deviation of the performance over 5 random seeds for a fair comparison. To make the results in figures clearer for readers, we adopt a 95% confidence interval to plot the error region.

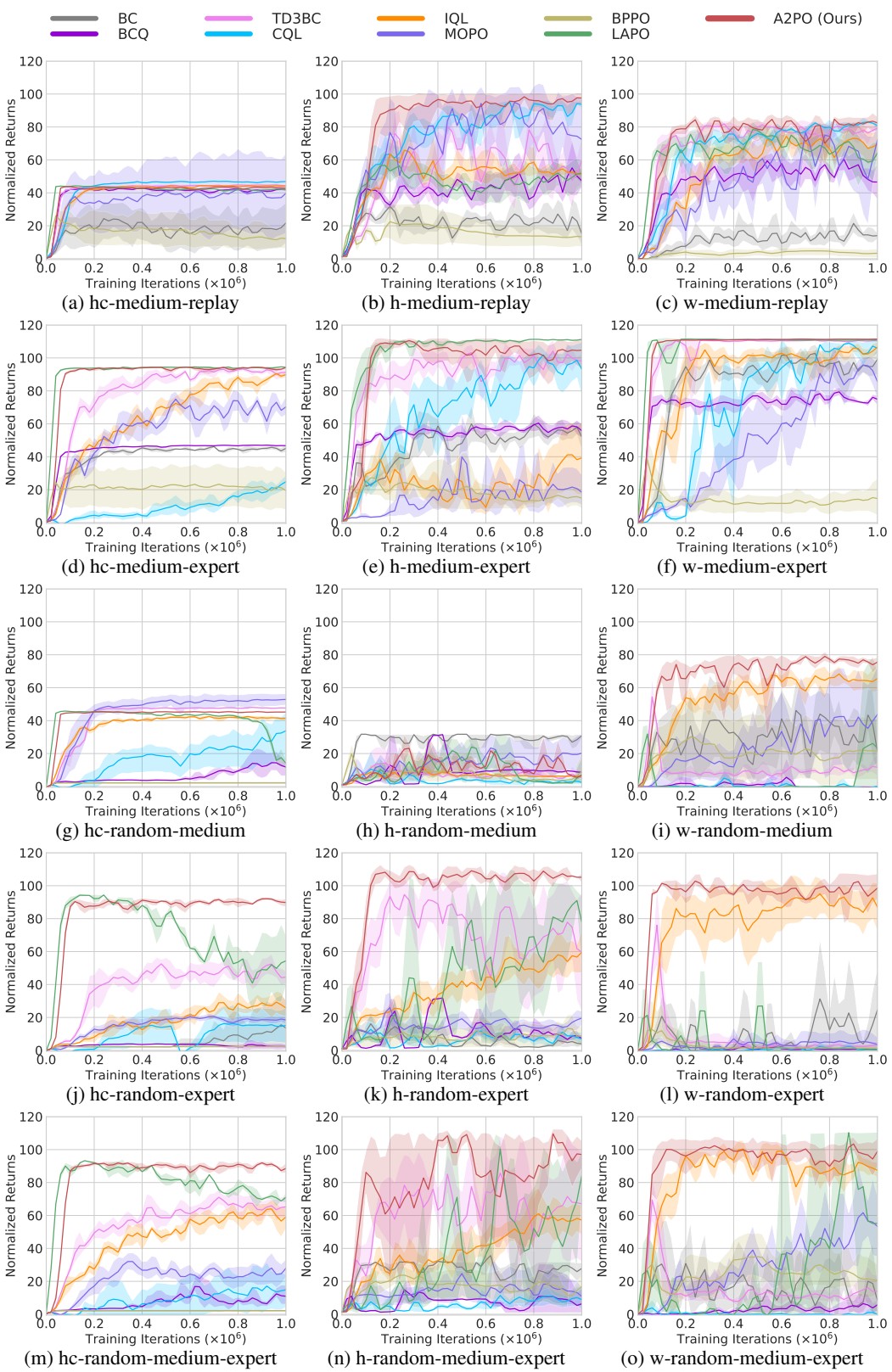

Figure S2: Learning curves of our proposed A2PO and baselines on the locomotion tasks under different multi-quality datasets.

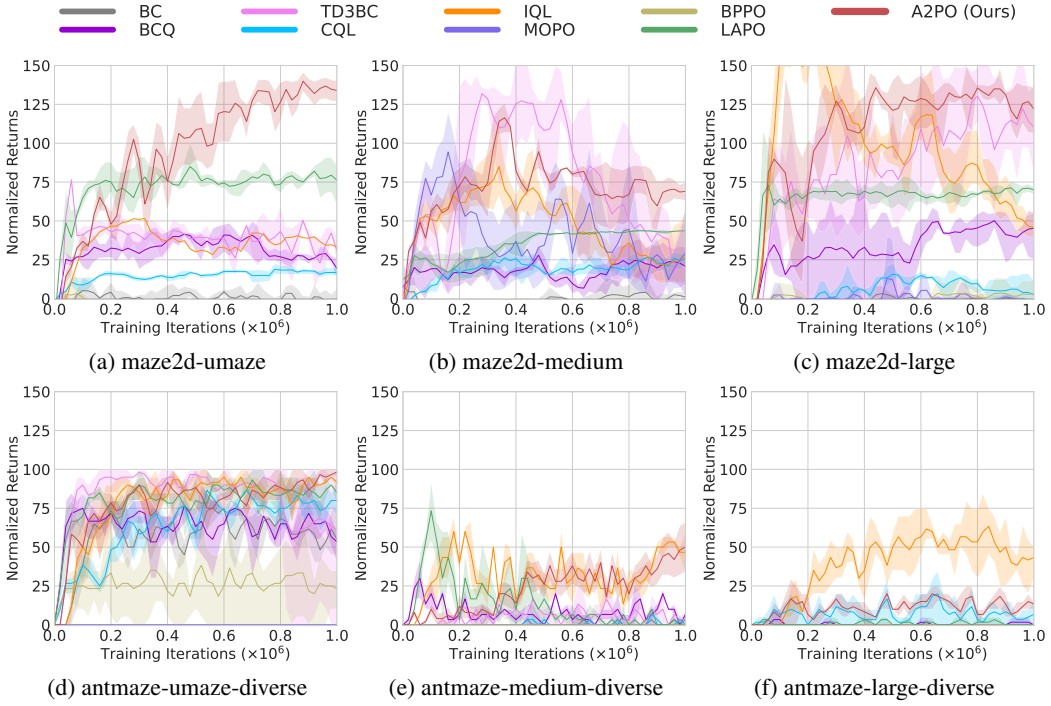

Figure S3: Learning curves of our proposed A2PO method and baselines on the navigation tasks under the single-quality *expert* dataset.

Table S5: Test returns of our proposed A2PO with different advantage conditions during training. $\pm$ corresponds to one standard deviation of the average evaluation of the performance on 5 random seeds. The performance is measured by the normalized scores at the last training iteration. **Bold** indicates the best performance in each task.

| Source | Task | $\xi_{\text{fix}} = 1$ | $\xi_{\text{dis}}, \epsilon = 0.0$ | $\xi_{\text{dis}}, \epsilon = 0.1$ | $\xi_{\text{dis}}, \epsilon = 0.5$ | Continuous $\xi$ |
|---|---|---|---|---|---|---|
| medium replay | halfcheetah | $39.63_{\pm0.56}$ | $40.67_{\pm0.81}$ | $40.96_{\pm0.49}$ | $41.24_{\pm0.50}$ | $\mathbf{44.74}_{\pm0.22}$ |
| | hopper | $74.81_{\pm11.21}$ | $96.92_{\pm1.70}$ | $85.19_{\pm5.99}$ | $93.33_{\pm4.36}$ | $\mathbf{101.59}_{\pm1.25}$ |
| | walker2d | $61.90_{\pm1.62}$ | $63.61_{\pm5.55}$ | $73.39_{\pm5.81}$ | $69.13_{\pm6.74}$ | $\mathbf{82.82}_{\pm1.70}$ |
| medium expert | halfcheetah | $93.10_{\pm1.54}$ | $94.11_{\pm0.39}$ | $94.46_{\pm0.49}$ | $94.77_{\pm0.04}$ | $\mathbf{95.61}_{\pm0.54}$ |
| | hopper | $62.50_{\pm5.54}$ | $\mathbf{110.45}_{\pm0.66}$ | $108.58_{\pm1.46}$ | $107.62_{\pm2.86}$ | $107.44_{\pm0.56}$ |
| | walker2d | $109.31_{\pm0.06}$ | $110.73_{\pm0.29}$ | $110.66_{\pm0.08}$ | $110.66_{\pm6.74}$ | $\mathbf{112.13}_{\pm0.24}$ |
| random expert | halfcheetah | $5.77_{\pm1.14}$ | $25.98_{\pm6.37}$ | $21.88_{\pm6.21}$ | $22.91_{\pm3.67}$ | $\mathbf{90.32}_{\pm1.63}$ |
| | hopper | $51.57_{\pm31.81}$ | $107.75_{\pm3.69}$ | $\mathbf{110.41}_{\pm1.00}$ | $95.84_{\pm19.41}$ | $105.19_{\pm4.54}$ |
| | walker2d | $63.05_{\pm33.08}$ | $73.64_{\pm51.98}$ | $\mathbf{109.64}_{\pm0.02}$ | $109.93_{\pm0.11}$ | $91.96_{\pm10.98}$ |
| random medium expert | halfcheetah | $63.16_{\pm1.17}$ | $73.67_{\pm5.36}$ | $71.57_{\pm2.65}$ | $71.18_{\pm3.65}$ | $\mathbf{90.58}_{\pm1.44}$ |
| | hopper | $65.74_{\pm38.22}$ | $108.03_{\pm1.16}$ | $96.12_{\pm15.79}$ | $\mathbf{108.81}_{\pm0.08}$ | $107.84_{\pm0.42}$ |
| | walker2d | $88.95_{\pm13.48}$ | $96.89_{\pm13.78}$ | $54.52_{\pm54.60}$ | $\mathbf{108.68}_{\pm3.33}$ | $97.71_{\pm6.74}$ |

Table S6: Test returns of A2PO with different discrete advantage conditions for test while using the original continuous advantage condition during training.

| Source | Task | $\pi_\omega(\cdot\|s, \xi=-1)$ | $\pi_\omega(\cdot\|s, \xi=0)$ | $\pi_\omega(\cdot\|s, \xi=1)$ |
|---|---|---|---|---|
| expert | halfcheetah | $87.15_{\pm0.27}$ | $93.23_{\pm0.38}$ | $\mathbf{96.26}_{\pm0.27}$ |
| | hopper | $84.87_{\pm22.18}$ | $106.56_{\pm6.66}$ | $\mathbf{111.70}_{\pm10.39}$ |
| | walker2d | $7.86_{\pm3.12}$ | $48.88_{\pm12.72}$ | $\mathbf{112.36}_{\pm0.23}$ |
| medium replay | halfcheetah | $4.74_{\pm5.56}$ | $21.51_{\pm8.57}$ | $\mathbf{44.74}_{\pm0.22}$ |
| | hopper | $11.44_{\pm6.32}$ | $25.28_{\pm1.08}$ | $\mathbf{101.59}_{\pm1.25}$ |
| | walker2d | $2.00_{\pm3.40}$ | $22.46_{\pm3.67}$ | $\mathbf{82.82}_{\pm1.70}$ |
| medium expert | halfcheetah | $40.42_{\pm0.64}$ | $64.45_{\pm4.63}$ | $\mathbf{95.61}_{\pm0.54}$ |
| | hopper | $37.07_{\pm16.15}$ | $76.85_{\pm24.41}$ | $\mathbf{107.44}_{\pm0.56}$ |
| | walker2d | $5.80_{\pm0.86}$ | $72.99_{\pm5.84}$ | $\mathbf{112.13}_{\pm0.24}$ |
| random expert | halfcheetah | $2.82_{\pm4.15}$ | $79.91_{\pm11.87}$ | $\mathbf{90.32}_{\pm1.63}$ |
| | hopper | $0.80_{\pm0.01}$ | $11.48_{\pm9.05}$ | $\mathbf{105.19}_{\pm4.54}$ |
| | walker2d | $-0.09_{\pm0.01}$ | $3.14_{\pm4.50}$ | $\mathbf{91.96}_{\pm10.98}$ |
| random medium expert | halfcheetah | $2.89_{\pm1.90}$ | $54.34_{\pm7.02}$ | $\mathbf{90.58}_{\pm1.44}$ |
| | hopper | $7.94_{\pm7.18}$ | $30.99_{\pm19.08}$ | $\mathbf{107.84}_{\pm0.42}$ |
| | walker2d | $1.54_{\pm1.62}$ | $9.16_{\pm5.91}$ | $\mathbf{97.71}_{\pm6.74}$ |

Table S7: Test returns of A2PO with different CVAE training steps (*i.e.*, the number of training iterations for CVAE optimization).

| Source | Task | $K = 1*10^5$ | $K = 1*10^6$ | $K = 2*10^5$ |
|---|---|---|---|---|
| medium replay | halfcheetah | $41.85_{\pm0.54}$ | $33.92_{\pm1.81}$ | $\mathbf{44.74}_{\pm0.22}$ |
| | hopper | $92.36_{\pm5.32}$ | $72.90_{\pm10.10}$ | $\mathbf{101.59}_{\pm1.25}$ |
| | walker2d | $78.03_{\pm2.95}$ | $43.22_{\pm5.73}$ | $\mathbf{82.82}_{\pm1.70}$ |
| medium expert | halfcheetah | $93.98_{\pm0.28}$ | $94.69_{\pm0.33}$ | $\mathbf{95.61}_{\pm0.54}$ |
| | hopper | $86.03_{\pm21.28}$ | $\mathbf{111.13}_{\pm0.45}$ | $107.44_{\pm0.56}$ |
| | walker2d | $111.41_{\pm0.33}$ | $111.14_{\pm0.17}$ | $\mathbf{112.13}_{\pm0.24}$ |
| random expert | halfcheetah | $74.72_{\pm8.36}$ | $1.45_{\pm0.11}$ | $\mathbf{90.32}_{\pm1.63}$ |
| | hopper | $\mathbf{105.78}_{\pm1.83}$ | $85.14_{\pm10.28}$ | $105.19_{\pm4.54}$ |
| | walker2d | $28.01_{\pm39.43}$ | $67.50_{\pm2.83}$ | $\mathbf{91.96}_{\pm10.98}$ |
| random medium expert | halfcheetah | $80.39_{\pm4.47}$ | $28.35_{\pm3.50}$ | $\mathbf{90.58}_{\pm1.44}$ |
| | hopper | $70.28_{\pm30.43}$ | $88.96_{\pm4.39}$ | $\mathbf{107.84}_{\pm0.42}$ |
| | walker2d | $\mathbf{99.90}_{\pm7.95}$ | $64.52_{\pm21.22}$ | $97.71_{\pm6.74}$ |

Table S8: Test returns of CVAE policy and agent policy in A2PO.

| Source | Task | CVAE policy $p_\psi(\cdot\|z_0, c^*)$ | Agent Policy $\pi(\cdot\|\tilde{z}^*, c^*)$ |
|---|---|---|---|
| random | halfcheetah | $15.31_{\pm 0.49}$ | $\mathbf{25.52}_{\pm 0.98}$ |
| | hopper | $\mathbf{31.66}_{\pm 0.00}$ | $18.43_{\pm 0.42}$ |
| | walker2d | $\mathbf{4.69}_{\pm 0.65}$ | $3.59_{\pm 1.74}$ |
| medium | halfcheetah | $45.73_{\pm 0.25}$ | $\mathbf{47.09}_{\pm 0.17}$ |
| | hopper | $57.06_{\pm 2.78}$ | $\mathbf{80.29}_{\pm 3.95}$ |
| | walker2d | $81.91_{\pm 0.70}$ | $\mathbf{84.88}_{\pm 0.23}$ |
| expert | halfcheetah | $94.95_{\pm 0.86}$ | $\mathbf{96.26}_{\pm 0.27}$ |
| | hopper | $91.85_{\pm 6.19}$ | $\mathbf{105.11}_{\pm 0.39}$ |
| | walker2d | $111.84_{\pm 0.52}$ | $\mathbf{112.36}_{\pm 0.23}$ |
| medium replay | halfcheetah | $39.17_{\pm 1.75}$ | $\mathbf{44.74}_{\pm 0.22}$ |
| | hopper | $91.47_{\pm 11.38}$ | $\mathbf{101.59}_{\pm 1.25}$ |
| | walker2d | $63.36_{\pm 9.49}$ | $\mathbf{82.82}_{\pm 1.70}$ |
| medium expert | halfcheetah | $93.35_{\pm 0.86}$ | $\mathbf{95.61}_{\pm 0.54}$ |
| | hopper | $\mathbf{112.20}_{\pm 0.56}$ | $107.44_{\pm 0.56}$ |
| | walker2d | $110.48_{\pm 0.28}$ | $\mathbf{112.13}_{\pm 0.24}$ |
| random medium | halfcheetah | $41.10_{\pm 0.89}$ | $\mathbf{45.20}_{\pm 0.21}$ |
| | hopper | $\mathbf{15.49}_{\pm 11.74}$ | $7.14_{\pm 0.35}$ |
| | walker2d | $41.89_{\pm 5.96}$ | $\mathbf{75.80}_{\pm 2.12}$ |
| random expert | halfcheetah | $36.89_{\pm 15.87}$ | $\mathbf{90.32}_{\pm 1.63}$ |
| | hopper | $81.38_{\pm 15.57}$ | $\mathbf{105.19}_{\pm 4.54}$ |
| | walker2d | $-0.06_{\pm 0.12}$ | $\mathbf{91.96}_{\pm 10.98}$ |
| random medium expert | halfcheetah | $66.19_{\pm 6.03}$ | $\mathbf{90.58}_{\pm 1.44}$ |
| | hopper | $56.67_{\pm 7.82}$ | $\mathbf{107.84}_{\pm 0.42}$ |
| | walker2d | $22.73_{\pm 6.05}$ | $\mathbf{97.71}_{\pm 6.74}$ |

## G  CVAE POLICY EVALUATION

In this section, we present thorough comparison results of the CVAE policy and agent policy in Table S8. The CVAE policy corresponds to the CVAE decoder $p_\psi(a|z_0, c^*)$, where $z_0$ is sampled from $\mathcal{N}(0, 1)$, state-advantage, $\xi^* = 1$ represents the largest advantage condition, and $c^* = s \,||\, \xi^*$. After $K$ steps of CVAE training is finished, the CVAE decoder $p_\psi(a|z_0, c^*)$ approximates the superior behavior policy output. The CVAE policy performance in Table S8 demonstrates that the CVAE policy only exhibits superior performance in a limited number of tasks and datasets, such as *hopper-random* and *walker2d-random*. In the majority of cases, the A2PO agent consistently outperforms the CVAE agent. These results indicate that the A2PO agent attains well-disentangled behavior policies and optimal agent policy, surpassing the capabilities of CVAE-reconstructed behavior policies.

## H  ADVANTAGE VISUALIZATION

In this section, we expand upon the didactic experiment introduced in Section 1 by incorporating additional tasks and datasets, which is aimed to indicate that the imprecise advantage approximation of AW methods is not a coincidence but a common problem. Similar to Figure 1, Figure S4 presents a comparative analysis of the actual return, LAPO, and our A2PO advantage approximation. The findings indicate that LAPO exhibits limited discrimination in assessing transition advantages, while our A2PO method effectively distinguishes between transitions of varying data quality. These results underscore the limitations of the AW method and highlight the superiority of our A2PO approach.

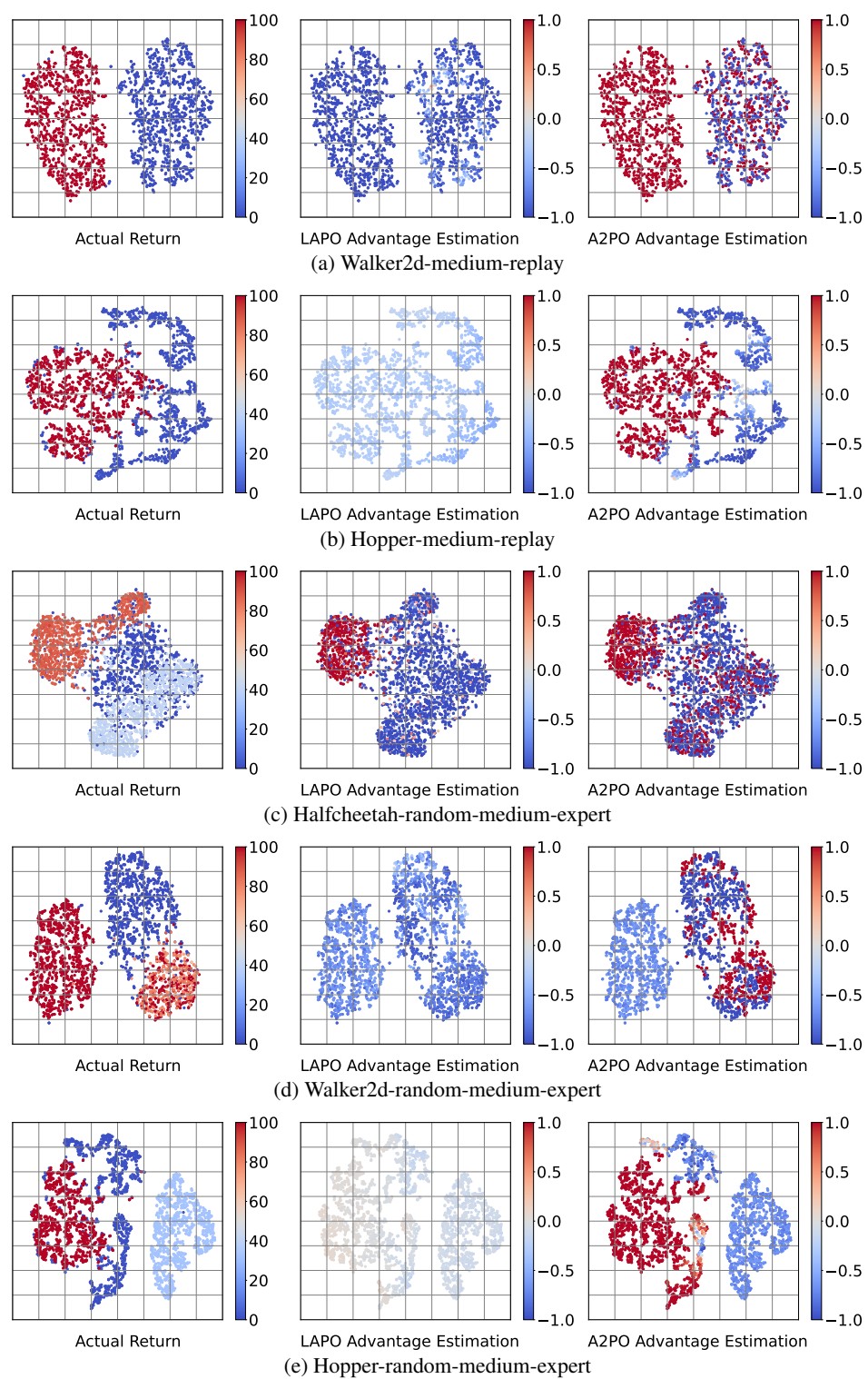

Figure S4: Comparison of our proposed A2PO method and the state-of-the-art AW method (LAPO) in advantage estimation for mixed-quality offline datasets in locomotion tasks. Each data point represents an initial state-action pair in the offline dataset after applying PCA, while varying shades of color indicate the magnitude of the actual return or advantage value.

## I  EXTRA COMPARISONS ON MORE TASKS

In this section, we test A2PO as well as other baselines in D4RL FrankaKitchen and Adroit tasks as shown in Table S9. The results indicate that our A2PO method can achieve comparable performance to the state-of-the-art lightweight baselines.

## J  EXTRA COMPARISONS WITH MORE BASELINES

In this section, we have added the lightweight baselines BEAR(Kumar et al., 2019), AWAC (Nair et al., 2020), SPOT (Wu et al., 2022), XQL (Garg et al., 2023), SQL (Xu et al., 2023), and EQL (Xu et al., 2023); representation learning baseline BPR (Zang et al., 2022); diffusing-based baselines Diffusion-QL (Wang et al., 2022) and IDQL (Hansen-Estruch et al., 2023) as the comparison baselines to further evaluate the superiority of our A2PO. The results are presented in Table S10 and Table S11. The results indicate that our A2PO method can achieve comparable performance to the state-of-the-art lightweight and representation learning baselines. Meanwhile, our A2PO exhibits performance on par with these more heavyweight diffusion-based methods. Notably, diffusion-based methods often perform poorly when there is a significant difference in data quality within the mixed-quality dataset. This observation underscores the effectiveness of our advantage-aware mechanism, which allows our lightweight CVAE model to capture the multi-quality characteristics from the offline dataset more effectively compared to the heavyweight diffusion models.

## K  ANALYSIS OF A2PO POLICY OPTIMIZATION

In this section, we consider the effectiveness of the BC regularization term in the A2PO policy optimization. In this case, A2PO and advantage-weighted method LAPO have the same policy optimization loss for a deterministic policy gradient. The comparison results are shown in Table S12. The results indicate that even without the BC regularization term, A2PO consistently outperforms LAPO in the majority of tasks. Moreover, the BC term in A2PO can enhance its performance in most cases. This comparison highlights the superior performance achieved by A2PO, showcasing its effective disentanglement of action distributions from different behavior policies in order to enforce a reasonable advantage-aware policy constraint and obtain an optimal agent policy.

## L  ANALYSIS OF THE PROPORTIONS OF THE MIXED-QUALITY DATASET

In this section, we have additionally conducted an ablation study to investigate the impact of varying amounts of single-quality samples in mixed-quality datasets, as shown in Table S13. The results demonstrate the robustness of our A2PO model in handling mixed-quality datasets containing different proportions of single-quality samples.

Table S9: Test returns of our proposed A2PO and baselines on the FrankaKitcdhen and Adroit tasks. *Italics* indicate that the results are obtained from the D4RL (Fu et al., 2020) paper.

| Task | BC | BCQ | CQL | IQL | TD3BC | SPOT | BPPO | LAPO | A2PO(Ours) |
|---|---|---|---|---|---|---|---|---|---|
| kitchen-complete-v0 | *33.8* | *8.1* | *43.8* | **62.5** | $0.83_{\pm1.18}$ | $41.67_{\pm11.40}$ | $0.00_{\pm0.00}$ | $50.83_{\pm10.27}$ | $60.00_{\pm2.04}$ |
| kitchen-partial-v0 | *33.8* | *18.9* | *49.8* | *46.3* | $0.00_{\pm0.00}$ | $0.00_{\pm0.00}$ | $16.67_{\pm2.36}$ | $\mathbf{58.75}_{\pm16.25}$ | $48.33_{\pm4.08}$ |
| kitchen-mixed-v0 | *47.5* | *10.6* | *51.0* | *51.0* | $0.83_{\pm1.18}$ | $0.00_{\pm0.00}$ | $0.00_{\pm0.00}$ | $52.33_{\pm6.27}$ | $\mathbf{53.33}_{\pm2.36}$ |
| pen-human-v1 | *34.4* | *12.3* | *37.5* | *71.5* | $-3.69_{\pm0.38}$ | $32.70_{\pm11.40}$ | $25.77_{\pm19.25}$ | $68.06_{\pm18.01}$ | $68.94_{\pm5.90}$ |
| hammer-human-v1 | *1.5* | *1.2* | ***4.4*** | *1.4* | $0.71_{\pm0.25}$ | $1.99_{\pm0.07}$ | $0.50_{\pm0.35}$ | $1.12_{\pm0.29}$ | $1.84_{\pm0.35}$ |
| door-human-v1 | *0.5* | *0.4* | ***9.9*** | *4.3* | $-0.33_{\pm0.01}$ | $-0.33_{\pm0.01}$ | $-0.02_{\pm0.03}$ | $6.07_{\pm4.60}$ | $8.51_{\pm3.66}$ |
| relocate-human-v1 | *0.0* | *0.0* | *0.2* | *0.1* | $-0.30_{\pm0.01}$ | $-0.07_{\pm0.13}$ | $-0.08_{\pm0.08}$ | $0.04_{\pm0.01}$ | $\mathbf{0.49}_{\pm0.58}$ |
| pen-cloned-v1 | *56.9* | *28.0* | *39.2* | *37.3* | $1.74_{\pm2.90}$ | $2.54_{\pm7.96}$ | $21.77_{\pm5.88}$ | $55.84_{\pm20.29}$ | $\mathbf{84.80}_{\pm21.43}$ |
| hammer-cloned-v1 | *0.8* | *0.4* | ***2.1*** | *2.1* | $0.26_{\pm0.02}$ | $0.46_{\pm0.03}$ | $0.25_{\pm0.22}$ | $0.84_{\pm0.35}$ | $0.43_{\pm0.15}$ |
| door-cloned-v1 | *-0.1* | *0.0* | *0.4* | ***1.6*** | $-0.35_{\pm0.02}$ | $-0.35_{\pm0.02}$ | $-0.04_{\pm0.05}$ | $0.22_{\pm0.40}$ | $0.31_{\pm0.04}$ |
| relocate-cloned-v1 | *-0.1* | *-0.2* | *-0.1* | *-0.2* | $-0.31_{\pm0.01}$ | $-0.30_{\pm0.01}$ | $-0.14_{\pm0.11}$ | $-0.05_{\pm0.07}$ | $\mathbf{0.00}_{\pm0.04}$ |
| Total | 209.0 | 79.7 | 238.2 | 277.9 | -0.61 | 78.31 | 64.68 | 294.05 | **326.98** |

Table S10: Test returns of our proposed A2PO and lightweight baselines on the locomotion tasks. The *italic* results of BEAR and AWAC are obtained from the D4RL paper, while SPOT, XQL, SQL and EQL are obtained from its original paper.

| Source | Task | BEAR | AWAC | SPOT | XQL | SQL | EQL | BPR | A2PO (Ours) |
|---|---|---|---|---|---|---|---|---|---|
| medium | halfcheetah | *41.7* | *43.5* | ***58.4*** | *48.3* | *48.3* | *47.2* | 47.25±0.34 | 47.09±0.17 |
| | hopper | *52.1* | *57.0* | ***86.0*** | *74.2* | *75.5* | *70.6* | 58.16±4.32 | 80.29±3.95 |
| | walker2d | *59.1* | *72.4* | ***86.4*** | *84.2* | *84.2* | *83.2* | 82.74±1.83 | 84.88±0.23 |
| medium replay | halfcheetah | *38.6* | *40.5* | ***52.2*** | *45.2* | *44.8* | *44.5* | 41.20±2.43 | 44.74±0.22 |
| | hopper | *19.2* | *37.2* | *100.2* | *100.7* | ***101.7*** | *98.1* | 41.86±7.98 | 101.59±1.25 |
| | walker2d | *33.7* | *27.0* | ***91.6*** | *82.2* | *77.2* | *81.6* | 83.30±25.11 | 82.82±1.70 |
| medium expert | halfcheetah | *53.4* | *42.8* | *86.9* | *94.2* | *94.0* | *94.6* | 95.16±0.94 | **95.61±0.54** |
| | hopper | *40.1* | *55.8* | *99.3* | ***111.2*** | *110.8* | *111.5* | 110.18±2.72 | 105.44±0.56 |
| | walker2d | *96.3* | *74.5* | *112.0* | ***112.7*** | *111.0* | *110.2* | 109.25±0.28 | 112.13±0.24 |
| random expert | halfcheetah | 4.18±1.79 | 87.32±2.91 | 60.42±3.69 | 35.68±9.07 | 30.60±12.35 | 47.37±6.41 | 3.10±1.70 | **90.32±1.63** |
| | hopper | 5.73±3.63 | 84.70±19.83 | 98.60±5.32 | 55.49±14.70 | 68.63±14.05 | 68.57±24.55 | 53.96±16.22 | **105.19±4.54** |
| | walker2d | -0.36±0.00 | 11.70±12.04 | 7.20±9.96 | 21.69±19.27 | **104.38±5.03** | 9.09±7.94 | 27.60±26.97 | 91.96±10.98 |
| random medium | halfcheetah | 13.78±4.87 | **46.54±0.10** | 46.12±0.25 | 39.71±2.76 | 36.93±3.82 | 42.32±1.45 | 39.86±1.37 | 45.20±0.21 |
| | hopper | 1.40±0.58 | 19.45±11.87 | **7.78±0.87** | 1.59±0.17 | 5.03±2.19 | 1.69±0.22 | 3.99±0.60 | 7.14±0.35 |
| | walker2d | -0.46±0.07 | -0.03±0.08 | 7.77±4.10 | 4.30±6.27 | 68.91±8.00 | 31.40±15.29 | 31.73±29.98 | **75.80±2.12** |
| random medium expert | halfcheetah | 2.25±0.00 | 89.65±1.67 | 80.18±7.47 | 42.73±12.32 | 63.42±6.37 | 42.79±3.18 | 26.40±11.76 | **90.58±1.44** |
| | hopper | 8.62±2.10 | 30.31±7.46 | 40.19±23.07 | 47.08±28.89 | 75.01±13.68 | 72.41±17.91 | 46.67±7.65 | **107.84±0.42** |
| | walker2d | -0.41±0.65 | -0.31±0.07 | 10.31±4.63 | 52.60±26.99 | 77.68±26.65 | 60.95±21.81 | 105.25±2.08 | **97.71±6.67** |
| Total | | 468.9 | 1205.23 | 1131.57 | 1053.77 | 1278.09 | 1118.09 | 1007.66 | **1466.33** |

Table S11: Test returns of our proposed A2PO, representation learning baseline BPR, and lightweight baselines on the locomotion tasks. The *italic* results of Diffusion-QL and IDQL are obtained from its original paper.

| Source | Task | Diffusion-QL | IDQL | A2PO (Ours) |
|---|---|---|---|---|
| medium | halfcheetah | *51.1* | ***51.0*** | 47.09±0.17 |
| | hopper | *90.5* | ***65.4*** | 80.29±3.95 |
| | walker2d | *87.0* | ***82.5*** | 84.88±0.23 |
| medium replay | halfcheetah | *47.8* | ***45.9*** | 44.74±0.22 |
| | hopper | *95.5* | *92.1* | **101.59±1.25** |
| | walker2d | *101.3* | ***85.1*** | 82.82±1.70 |
| medium expert | halfcheetah | *96.8* | ***95.9*** | 95.61±0.54 |
| | hopper | ***111.1*** | *108.6* | 105.44±0.56 |
| | walker2d | *110.1* | ***112.7*** | 112.13±0.24 |
| random expert | halfcheetah | 86.07±1.49 | 32.55±2.94 | **90.32±1.63** |
| | hopper | 101.96±7.03 | 19.73±13.80 | **105.19±4.54** |
| | walker2d | 56.33±31.44 | 0.18±0.08 | **91.96±10.98** |
| random medium | halfcheetah | **48.43±0.32** | 6.26±1.18 | 45.20±0.21 |
| | hopper | 6.93±0.75 | 2.78±2.01 | **7.14±0.35** |
| | walker2d | 3.27±2.35 | 3.82±3.21 | **75.80±2.12** |
| random medium expert | halfcheetah | 81.15±7.21 | 36.24±13.27 | **90.58±1.44** |
| | hopper | 70.09±2.05 | 6.17±3.31 | **107.84±0.42** |
| | walker2d | 56.56±23.02 | 18.55±8.13 | **97.71±6.74** |
| Total | | 1301.99 | 865.48 | **1466.33** |

Table S12: Test returns of LAPO and our proposed A2PO without BC term in the policy optimization step. **Bold** indicates that the better performance among LAPO and A2PO w/o BC.

| Source | Task | LAPO | A2PO w/o BC | A2PO (Ours) |
|---|---|---|---|---|
| medium | halfcheetah | $45.58_{\pm0.06}$ | $\mathbf{46.81}_{\pm0.09}$ | $47.09_{\pm0.17}$ |
| | hopper | $52.53_{\pm2.61}$ | $\mathbf{70.08}_{\pm4.03}$ | $80.29_{\pm3.95}$ |
| | walker2d | $80.46_{\pm1.25}$ | $\mathbf{81.97}_{\pm1.12}$ | $84.88_{\pm0.23}$ |
| medium replay | halfcheetah | $41.94_{\pm0.47}$ | $\mathbf{42.01}_{\pm0.28}$ | $44.74_{\pm0.22}$ |
| | hopper | $50.14_{\pm11.16}$ | $\mathbf{96.51}_{\pm1.47}$ | $101.59_{\pm1.25}$ |
| | walker2d | $60.55_{\pm10.45}$ | $\mathbf{71.09}_{\pm7.98}$ | $82.82_{\pm1.70}$ |
| medium expert | halfcheetah | $94.22_{\pm0.46}$ | $\mathbf{94.29}_{\pm0.03}$ | $95.61_{\pm0.54}$ |
| | hopper | $\mathbf{111.04}_{\pm0.36}$ | $107.27_{\pm1.93}$ | $105.44_{\pm0.56}$ |
| | walker2d | $110.88_{\pm0.15}$ | $\mathbf{111.61}_{\pm0.07}$ | $112.13_{\pm0.24}$ |
| random expert | halfcheetah | $\mathbf{52.58}_{\pm17.30}$ | $31.37_{\pm6.27}$ | $90.32_{\pm1.63}$ |
| | hopper | $82.33_{\pm18.95}$ | $\mathbf{113.20}_{\pm1.20}$ | $105.19_{\pm4.54}$ |
| | walker2d | $0.89_{\pm0.53}$ | $\mathbf{66.82}_{\pm11.01}$ | $91.96_{\pm10.98}$ |
| random medium | halfcheetah | $18.53_{\pm0.99}$ | $\mathbf{43.19}_{\pm0.54}$ | $45.20_{\pm0.21}$ |
| | hopper | $\mathbf{4.17}_{\pm3.11}$ | $1.59_{\pm0.92}$ | $7.14_{\pm0.35}$ |
| | walker2d | $23.65_{\pm33.97}$ | $\mathbf{72.32}_{\pm4.43}$ | $75.80_{\pm2.12}$ |
| random medium expert | halfcheetah | $\mathbf{71.09}_{\pm0.47}$ | $70.77_{\pm4.21}$ | $90.58_{\pm1.44}$ |
| | hopper | $66.59_{\pm19.29}$ | $\mathbf{86.54}_{\pm7.25}$ | $107.84_{\pm0.42}$ |
| | walker2d | $60.41_{\pm43.22}$ | $\mathbf{110.35}_{\pm1.20}$ | $97.71_{\pm6.74}$ |
| Total | | 1027.58 | **1317.79** | 1466.33 |

Table S13: Test returns of our proposed A2PO on the locomotion tasks with the medium-expert (m-e) dataset containing different proportions of single-quality samples.

| Task | m:e = 3:1 | m:e = 2:1 | m:e = 1:1 | m:e = 1:2 | m:e = 1:3 |
|---|---|---|---|---|---|
| halfcheetah | $93.78_{\pm0.70}$ | $94.28_{\pm0.09}$ | $\mathbf{95.61}_{\pm0.54}$ | $95.30_{\pm0.25}$ | $95.06_{\pm0.22}$ |
| hopper | $106.94_{\pm0.67}$ | $74.72_{\pm25.18}$ | $105.44_{\pm0.56}$ | $\mathbf{112.19}_{\pm0.09}$ | $110.17_{\pm1.37}$ |
| walker2d | $110.94_{\pm0.36}$ | $110.77_{\pm0.57}$ | $\mathbf{112.13}_{\pm0.24}$ | $111.26_{\pm0.49}$ | $110.67_{\pm0.21}$ |
| Total | 311.66 | 279.77 | 313.18 | **318.75** | 315.90 |

