# OpenReview forum: "Advantage-Aware Policy Optimization for Offline Reinforcement Learning"
_ICLR.cc/2024/Conference — Submitted to ICLR 2024_

### Official Review · Reviewer_HeZ6 · 2023-10-25

**Soundness:** 2 fair
**Presentation:** 2 fair
**Contribution:** 2 fair
**Rating:** 3
**Confidence:** 4

**Summary:**

This paper proposes an offline RL algorithm which uses CVAE to map the actions into the latent space and then perform latent space TD3+BC for policy optimization. The key design as claimed by the authors is to introduce the projected advantage values as an additional conditional input in CVAE, which can help better distinguish data quality. The proposed method lacks concrete theoretical backing on why it will work, and seems to require careful hyperparameter tuning to maintain stability and achieve good performance. See the following strengths and weaknesses for detailed comments.

**Strengths:**

- The paper is easy to read.
- Relatively comprehensive evaluations on MuJoCo and antmaze tasks. Reasonable performance. Though I find some antmaze task scores seem to be problematic.

**Weaknesses:**

- The design of adding advantage values into a jointly trained CVAE can lead to extra training instability. As the value functions are gradually learned, the advantage value can keep changing during the CVAE training. This means that the conditioning information in CVAE is not stationary, causing the learning of CVAE to also fluctuate. It can become worse when using the latent representation from a non-stationary CVAE, which can further cause instability in the value and policy learning. This is evident as the authors have admitted that the training step $K$ of CVAE needs to be carefully tuned, and as shown in Fig. 5, the proposed method can fail miserably if using an improper $K$ value.
- The references on offline RL algorithms are mostly restricted to those before 2022. As this field is rapidly developing, there are many strong offline RL algorithms have been proposed in 2023 but lack acknowledgment or comparison. The reported scores are only comparable to some of the newer but more lightweight in-sample learning algorithms like XQL[1] and SQL[2], and less performant than some recent methods that use diffusion policies [3, 4]. Given the proposed method is somewhat heavy (needs to learn an additional, potentially unstable CVAE), I don't think the proposed method offers impressive performance.
- The value and policy learning procedure is essentially TD3+BC[5], but conducted in the latent space. However, TD3+BC is never mentioned in Section 4.2. Even in the codes provided by the author, the corresponding method names are "TD3BC_critic_loss" and "TD3BC_actor_loss". Same as TD3+BC, the actual implementation needs to use a $\lambda$ hyperparameter to trade off max Q and minimize the BC penalty. However, this is not mentioned in Eq.(8). This extra hyperparameter is intentionally omitted and corresponding hyperparameter tuning is not discussed. If this hyperparameter is carefully tuned for each task, even the original TD3+BC is likely to have strong performance.
- I feel the proposed method is essentially doing some kind of representation learning over TD3+BC, but unlike other representation learning methods, it could be more unstable. It is suggested that the proposed method should also be compared with existing offline RL baselines with representation learning.
- In the experiments, adding additional experiments on datasets mixed with random data is not very meaningful for practical settings. As nobody will use bad random data to train their offline RL policy in practice. Moreover, BPPO is a flawed offline RL algorithm and is not worth comparison. In its code implementation, it hides online evaluation during offline training, which is a kind of cheating in the offline RL setting. It is a shame that such a paper even gets accepted in ICLR 2023. On the other hand, some recent strong and rigorous offline RL algorithms should be compared.

**References:**

[1] Garg, D., Hejna, J., Geist, M., & Ermon, S. Extreme Q-Learning: MaxEnt RL without Entropy. ICLR 2023.

[2] Xu, H., et al. Offline RL with no OOD actions: In-sample learning via implicit value regularization. ICLR 2023.

[3] Wang, Z. et al. Diffusion policies as an expressive policy class for offline reinforcement learning. ICLR 2023.

[4] Hansen-Estruch, P. et al. Idql: Implicit q-learning as an actor-critic method with diffusion policies. arXiv preprint arXiv:2304.10573, 2023.

[5] Fujimoto, S., & Gu, S. S. A minimalist approach to offline reinforcement learning. NeurIPS 2021.

**Questions:**

- The scores for antmaze-m-d and antmze-l-d look strange, CQL and IQL are reported to have reasonable scores in the IQL paper as well as in previous references [1-2,4], however, their scores are all zero in this paper. Moreover, the proposed A2PO's performance looks very bad, inferior to scores of baselines as reported in other papers. Why is that?
- Please report the max-Q and BC trade-off hyperparameter values in the TD3+BC style policy improvement step (Eq.(8)). Has it been tuned for different tasks?

---

> ### Author Response · Authors · 2023-11-19
> **Author Response (Part 1/4)**
>
> We greatly appreciate the reviewer for the insightful comments, which help us improve the quality of the paper significantly. Below we address the main points raised in the review. We will update the additional experimental results in the revision later.
>
>
>
> **[Weakness 1]: The design of adding advantage values into a jointly trained CVAE can lead to extra training instability.**
>
>
> We agree that introducing an additional network may lead to instability, but we believe it is worthwhile for the enhancement in performance. The experimental results show that our A2PO method exhibits superior performance, particularly in mixed-quality datasets. Moreover, regarding the hyperparameter of CVAE training steps, our A2PO can also demonstrate promising robustness. The number of training steps is a crucial hyperparameter for any deep learning model and is not unique to our proposed CVAE. A lower number of training steps may result in non-convergence of the model, whereas a higher number can lead to overfitting. The ablation study conducted in our paper actually involved selecting training steps at two extreme ends, which is why there was a drastic drop in performance. As shown in Table S4 of our original paper, except for the specific requirements in the antmaze and maze2d environments, a value of $K=2*10^5$ proves to be sufficient for all other scenarios. This highlights the robustness of the CVAE training step across various tasks and datasets.
>
>
> **[Weakness 2]: I don't think the proposed method offers impressive performance.**
>
> We have additionally compared our A2PO to the mentioned lightweight baselines, including XQL, SQL, and EQL, and the diffusion-based policy baselines, including IDQL and Diffusion-QL. The corresponding results are shown in Table R1 and Table R2.
>
> The results in Table R1 indicate that our A2PO method can achieve comparable performance to the state-of-the-art lightweight baselines. The results in Table R2 show that our A2PO exhibits performance on par with these more heavyweight diffusion-based methods. Notably, diffusion-based methods often perform poorly when there is a significant difference in data quality within the mixed-quality dataset. This observation underscores the effectiveness of our advantage-aware mechanism, which allows our lightweight CVAE model to capture the multi-modal characteristics from the offline dataset more effectively compared to the heavyweight diffusion models.
>
> While the diffusion-based methods solely rely on the diffusion models to implicitly learn the distribution of offline data, our advantage-aware mechanism explicitly disentangles the action distributions of intertwined behavior policies by modeling the advantage values of all training data as conditional variables. It is noted that these two approaches are orthogonal. This implies that our advantage-aware mechanism can be effectively integrated with the diffusion-based methods by introducing the conditional diffusion models [1], thereby enhancing their capacity to process multi-modal data. This is an interesting direction for future work.

---

> ### Author Response · Authors · 2023-11-19
> **Author Response (Part 2/4)**
>
> Table R1. Test returns of our A2PO and new state-of-the-art lightweight baselines. *Italics* indicates that the results are obtained from the original paper, while the other results are evaluated over 3 random seeds based on the corresponding official code.
> | Env                                 | XQL               | SQL                   | EQL               | A2PO                  |
> | ----------------------------------- | ----------------- | --------------------- | ----------------- | --------------------- |
> | halfcheetah-medium-v2               | ***48.3***        | ***48.3***            | *47.2*            | 47.09 $\pm$ 0.17      |
> | hopper-medium-v2     | *74.2*            | *75.5*                | *70.6*            | **80.29 $\pm$ 3.95**  |
> | walker2d-medium-v2                  | *84.2*            | *84.2*                | *83.2*            | **84.88 $\pm$ 0.23**  |
> | halfcheetah-medium-replay-v2        | *45.2*            | ***44.8***            | *44.5*            | 44.74 $\pm$ 0.22      |
> | hopper-medium-replay-v2             | *100.7*           | ***101.7***           | *98.1*            | 101.59 $\pm$ 1.25     |
> | walker2d-medium-replay-v2           | *82.2*            | *77.2*                | *81.6*            | **82.82 $\pm$ 1.70**  |
> | halfcheetah-medium-expert-v2        | *94.2*            | *94.0*                | *94.6*            | **95.61 $\pm$ 0.54**  |
> | hopper-medium-expert-v2             | 111.2             | *110.8*               | ***111.5***       | 105.44 $\pm$ 0.56     |
> | walker2d-medium-expert-v2           | ***112.7***       | *111.0*               | *110.2*           | 112.13 $\pm$ 0.24     |
> | halfcheetah-random-expert-v2        | 35.68 $\pm$ 9.07  | 30.60 $\pm$ 12.35     | 47.37 $\pm$ 6.41  | **90.32 $\pm$ 1.63**  |
> | hopper-random-expert-v2             | 55.49 $\pm$ 14.7  | 68.63 $\pm$ 14.05     | 68.57 $\pm$ 24.55 | **105.19 $\pm$ 4.54** |
> | walker2d-random-expert-v2 | 21.69 $\pm$ 19.27 | **104.38 $\pm$ 5.03** | 9.09 $\pm$ 7.94   | 91.96 $\pm$ 10.98     |
> | halfcheetah-random-medium-v2   | 39.71 $\pm$ 2.76  | 36.93 $\pm$ 3.82      | 42.32 $\pm$ 1.45  | **45.20 $\pm$ 0.21**  |
> | hopper-random-medium-v2    | 1.59 $\pm$ 0.17   | 5.03 $\pm$ 2.19       | 1.69 $\pm$ 0.22   | **7.14 $\pm$ 0.35**   |
> | walker2d-random-medium-v2  | 4.30 $\pm$ 6.27   | 68.91 $\pm$ 8.0       | 31.40 $\pm$ 15.29 | **75.80 $\pm$ 2.12**  |
> | halfcheetah-random-medium-expert-v2 | 42.73 $\pm$ 12.32 | 63.42 $\pm$ 6.37      | 42.79 $\pm$ 3.18  | **90.58 $\pm$ 1.44**  |
> | hopper-random-medium-expert-v2      | 47.08 $\pm$ 28.89 | 75.01 $\pm$ 13.68     | 72.41 $\pm$ 17.91 | **107.84 $\pm$ 0.42** |
> | walker2d-random-medium-expert-v2    | 52.60 $\pm$ 26.99 | 77.68 $\pm$ 26.65     | 60.95 $\pm$ 21.81 | **97.71 $\pm$ 6.74**  |
> | **Total**   | 1053.77 | 1278.09 | 1118.09 | **1466.33**  |
>
>
>
>
>
> Table R2. Test returns of our A2PO and diffusion-based baselines.
> | Env                                 | Diffusion-QL         | IDQL              | A2PO                  |
> | ----------------------------------- | -------------------- | ----------------- | --------------------- |
> | halfcheetah-medium-v2      | ***51.1***           | *51*              | 47.09 $\pm$ 0.17      |
> | hopper-medium-v2   | ***90.5***           | *65.4*            | 80.29 $\pm$ 3.95      |
> | walker2d-medium-v2 | ***87.0***           | *82.5*            | 84.88 $\pm$ 0.23      |
> | halfcheetah-medium-replay-v2        | ***47.8***           | *45.9*            | 44.74 $\pm$ 0.22      |
> | hopper-medium-replay-v2      | *95.5*               | *92.1*            | **101.59 $\pm$ 1.25** |
> | walker2d-medium-replay-v2           | ***101.3***          | *85.1*            | 82.82 $\pm$ 1.70      |
> | halfcheetah-medium-expert-v2        | ***96.8***           | *95.9*            | 95.61 $\pm$ 0.54      |
> | hopper-medium-expert-v2      | ***111.1***          | *108.6*           | 105.44 $\pm$ 0.56     |
> | walker2d-medium-expert-v2      | *110.1*              | ***112.7***       | 112.13 $\pm$ 0.24     |
> | halfcheetah-random-expert-v2   | 86.07 $\pm$ 1.49     | 32.55 $\pm$ 2.94  | **90.32 $\pm$ 1.63**  |
> | hopper-random-expert-v2  | 101.96 $\pm$ 7.03    | 19.73 $\pm$ 13.80 | **105.19 $\pm$ 4.54** |
> | walker2d-random-expert-v2   | 56.33 $\pm$ 31.44    | 0.18 $\pm$ 0.08   | **91.96 $\pm$ 10.98** |
> | halfcheetah-random-medium-v2   | **48.43 $\pm$ 0.32** | 6.26 $\pm$ 1.18   | 45.20 $\pm$ 0.21      |
> | hopper-random-medium-v2   | 6.93 $\pm$ 0.75      | 2.78 $\pm$ 2.01   | **7.14 $\pm$ 0.35**   |
> | walker2d-random-medium-v2   | 3.27 $\pm$ 2.35      | 3.82 $\pm$ 3.21   | **75.80 $\pm$ 2.12**  |
> | halfcheetah-random-medium-expert-v2 | 81.15 $\pm$ 7.21     | 36.24 $\pm$ 13.27 | **90.58 $\pm$ 1.44**  |
> | hopper-random-medium-expert-v2   | 70.09 $\pm$ 2.05     | 6.17 $\pm$ 3.31   | **107.84 $\pm$ 0.42** |
> | walker2d-random-medium-expert-v2    | 56.56 $\pm$ 23.02    | 18.55 $\pm$ 8.13  | **97.71 $\pm$ 6.74**  |
> | **Total**    | 1301.99    | 865.48  | **1466.33**  |

---

> ### Author Response · Authors · 2023-11-19
> **Author Response (Part 3/4)**
>
> **[Weakness 3 \& Question 2]: Please report the max-Q and BC trade-off hyperparameter values in the TD3+BC style policy improvement step (Eq.(8)). Has it been tuned for different tasks?**
>
> Thanks for the suggestions. As stated in Appendix A.2 (Implementation Details) of our original manuscript, we have illustrated that our implementation follows the TD3BC approach. We follow the same trade-off hyperparameter value as in the official TD3BC implementation and keep this hyperparameter on the whole experiment without tuning.
>
> As suggested by the reviewer, we have added more details about TD3BC in the revision:
>
> (1) In Section 4.2 (Agent Policy Optimazation): "For the policy improvement step, the TD3BC-style loss is defined as:..., following TD3BC, we add a normalization coefficient $\lambda={\alpha}/{(\frac{1}{N}\sum\_{(s\_i, a\_i)} |Q(s\_i, a\_i)|)}$ to the first term to keep the scale balance between Q value objective and regularization, where $\alpha$ is a hyperparameter to control the scale of the normalized Q value."
>
> (2) In Appendix B.2 (Implementation Details): "For the hyperparameter value $\alpha$ in Q normalization, we follow the same setting $\alpha=2.5$ as in the official TD3BC implementation and keep this hyperparameter on the whole experiment."
>
> **[Weakness 4]: It is suggested that the proposed method should also be compared with existing offline RL baselines with representation learning.**
>
> Thanks for the suggestion. We have additionally compared our A2PO with offline RL baseline with representation learning, BPR [2]. The results show that our A2PO can demonstrate promising results.
>
> Table R3. Test returns of our proposed A2PO and BPR.
> | Env                                 | BPR                   | A2PO                  |
> | ----------------------------------- | --------------------- | --------------------- |
> | halfcheetah-medium-v2               | **47.25 $\pm$ 0.34**  | 47.09 $\pm$ 0.17      |
> | hopper-medium-v2                    | 58.16 $\pm$ 4.32      | **80.29 $\pm$ 3.95**  |
> | walker2d-medium-v2                  | 82.74 $\pm$ 1.83      | **84.88 $\pm$ 0.23**  |
> | halfcheetah-medium-replay-v2        | 41.20 $\pm$ 2.43      | **44.74 $\pm$ 0.22**  |
> | hopper-medium-replay-v2             | 41.86 $\pm$ 7.98      | **101.59 $\pm$ 1.25** |
> | walker2d-medium-replay-v2           | 83.30 $\pm$ 25.11     | **82.82 $\pm$ 1.70**  |
> | halfcheetah-medium-expert-v2        | 95.16 $\pm$ 0.94      | **95.61 $\pm$ 0.54**  |
> | hopper-medium-expert-v2             | **110.18 $\pm$ 2.72** | 105.44 $\pm$ 0.56     |
> | walker2d-medium-expert-v2           | 109.25 $\pm$ 0.28     | **112.13 $\pm$ 0.24** |
> | halfcheetah-random-expert-v2        | 3.10 $\pm$ 1.07       | **90.32 $\pm$ 1.63**  |
> | hopper-random-expert-v2             | 53.96 $\pm$ 16.22     | **105.19 $\pm$ 4.54** |
> | walker2d-random-expert-v2           | 27.60 $\pm$ 26.97     | **91.96 $\pm$ 10.98** |
> | halfcheetah-random-medium-v2        | 39.86 $\pm$ 1.37      | **45.20 $\pm$ 0.21**  |
> | hopper-random-medium-v2             | 3.99 $\pm$ 0.60       | **7.14 $\pm$ 0.35**   |
> | walker2d-random-medium-v2           | 31.73 $\pm$ 29.98     | **75.80 $\pm$ 2.12**  |
> | halfcheetah-random-medium-expert-v2 | 26.40 $\pm$ 11.76     | **90.58 $\pm$ 1.44**  |
> | hopper-random-medium-expert-v2      | 46.67 $\pm$ 7.65      | **107.84 $\pm$ 0.42** |
> | walker2d-random-medium-expert-v2    | **105.25 $\pm$ 2.08** | 97.71 $\pm$ 6.74      |
> | **Total**    | 1007.66 |  **1466.33**    |

---

> ### Author Response · Authors · 2023-11-19
> **Author Response (Part 4/4)**
>
> **[Weakness 5]: (1) Experiments on datasets mixed with random data is not very meaningful for practical settings. (2) BPPO is a flawed offline RL algorithm and is not worth comparison. (3) Some recent strong and rigorous offline RL algorithms should be compared.**
>
>
> (1) We would like to point out that conducting experiments under various dataset settings is of great significance, as the quality of the dataset is often diverse in many practical scenarios. Random data has a low acquisition cost and also contains some exploration information. The Offline RL technique should be able to utilize datasets of different qualities to improve the agent performance as much as possible instead of only considering high-quality data. Moreover, our A2PO method can leverage arbitrary transition samples to estimate the advantage value for training. This eliminates the need for explicit trajectory return signals, rendering our approach more adaptable and practical for real-world applications where only transition data is accessible and the returns of entire trajectories are not known.
>
> (2) We compare A2PO with advantage-weighted BPPO because there is limited existing research on advantage-based methods. We also agree that it is unreasonable that the official implementation of BPPO uses online evaluation for offline BC model selection. For a fair comparison, in our original reproduction experiment, we have removed online evaluation for offline BC model selection in BPPO. Thus, the reported results in our original manuscript are convincing.
>
> (3) We have added more lightweight baselines (XQL [3], EQL [6], SQL [6]), Diffusion-policy baselines (Diffusion-QL [4], IDQL [7]), and representation learning baseline (BPR [2]) for a much more rigorous comparison as mentioned above. Compared to the additional baselines, our A2PO demonstrates promising results.
>
> **[Question 1]: The scores for antmaze-m-d and antmze-l-d look strange.**
>
> Sorry for the confusion. The discrepancy in scores arises from that we perform training in the maze2d and antmaze tasks without proper hyperparameters as in the official papers. Specifically, for CQL we follow the default setting of Lagrange threshold $\tau=1.0$ for locomotion tasks instead of $\tau=5.0$ for maze2d/antmaze navigation tasks. For IQL we follow the default setting of expectile $\tau=0.9$ and temperature $\beta=10.0$ for locomotion tasks instead of $\tau=0.7$ and $\beta=3.0$ for maze2d and antmaze navigation tasks. We have corrected these settings and performed new experiments. The results in Table R4 show that compared to CQL and IQL with the official settings, our A2PO can still achieve superior performance. We have corrected these results in our revision.  Moreover, we use "v1" version for experiments in the antmaze and maze2d tasks as in LAPO [3], while many of the other baselines use "v0" version (XQL [4], Diffusion-QL [5], D4RL [6]) or "v2" version (EQL [7], SQL [7], IDQL [8]) for experiments. Note that performance is not comparable across task versions.
>
>
> Table R4. Test returns of CQL and IQL following the hyperparameters in their official papers.
>
>
> | Env               | CQL              | IQL                   | A2PO                  |
> | ------------------------- | ---------------- | --------------------- | --------------------- |
> | maze2d-umaze-v1           | 17.02 $\pm$ 1.87 | 56.17 $\pm$ 9.86      | **133.27 $\pm$ 9.58** |
> | maze2d-medium-v1          | 22.45 $\pm$ 6.70 | 25.67 $\pm$ 16.93     | **83.95 $\pm$ 10.56** |
> | maze2d-large-v1           | 2.53 $\pm$ 6.58  | 45.67 $\pm$ 18.91     | **127.61 $\pm$ 5.35** |
> | antmaze-umaze-diverse-v1  | 80.00 $\pm$ 8.16 | 86.67 $\pm$ 12.47     | **96.66 $\pm$ 4.71**  |
> | antmaze-medium-diverse-v1 | 0.00 $\pm$ 0.00  | 46.67 $\pm$ 18.86     | **50.00 $\pm$ 15.25** |
> | antmaze-large-diverse-v1  | 6.67 $\pm$ 4.71  | **43.33 $\pm$ 12.47** | 6.16 $\pm$ 4.90       |
> | **Total** | 128.67  | 304.18 | **497.49**      |
>
>
>
>
> **References**
>
> [1] Ho, J., & Salimans, T. Classifier-free diffusion guidance. arXiv 2022.
>
> [2] Zhang, H. et al. Behavior Prior Representation Learning for Offline Reinforcement Learning. ICLR 2023.
>
> [3] Chen, X. et al. LAPO: Latent-Variable Advantage-Weighted Policy Optimization for Offline Reinforcement Learning. NeurIPS 2022.
>
> [4] Garg, D., Hejna, J., Geist, M., & Ermon, S. Extreme Q-Learning: MaxEnt RL without Entropy. ICLR 2023.
>
> [5] Wang, Z. et al. Diffusion policies as an expressive policy class for offline reinforcement learning. ICLR 2023.
>
> [6] Fu, J. et al. D4RL: Datasets for Deep Data-Driven Reinforcement Learning. arXiv 2020.
>
> [7] Xu, H., et al. Offline RL with no OOD actions: In-sample learning via implicit value regularization. ICLR 2023.
>
> [8] Hansen-Estruch, P. et al. Idql: Implicit q-learning as an actor-critic method with diffusion policies. arXiv 2023.

---

> > ### Comment · Reviewer_HeZ6 · 2023-11-20
> > **Thanks for your responses**
> >
> > I'd like to thank the authors for their responses and additional experiments. I've read the responses from the authors, as well as comments from other reviewers. However, I still hold a generally negative impression of this paper. The reasons are:
> > - I still have concerns regarding the potential instability caused by the jointly trained CVAE, which is conditioned on varying advantage values. This could increase the burden for hyperparameter tuning, which is extremely undesirable for offline RL practical settings.
> > - I still think that the method does not offer impressive performance given its relatively heavy architecture and extra hyperparameters. The performances of the proposed method in many cases are inferior to diffusion-based methods, and are comparable to more lightweight in-sample learning algorithms. The proposed method performs well in its newly introduced random-medium/expert datasets. However, these datasets do not belong to the standard D4RL test suites. As I mentioned in my previous review, in practice, such dataset compositions are not very meaningful, as no one will combine bad random data with expert data to train their offline RL policies. Moreover, in many real-world offline RL applications, the offline data are collected from the systems' historical logs, there are **NO** random data in these systems. I disagree with the authors that "random data has a low acquisition cost". If you collect such data using random policy in many mission-critical real-world systems, there could be disastrous consequences. Hence I think adding scores of these random-mixed datasets to the total score does not necessarily demonstrate the superiority of the proposed method in practice.
> > - Regarding the antmaze tasks: What is the A2PO's performance on antmaze-v0 or v2 datasets? Since most existing offline RL papers report their scores in antmaze-v0 or v2 tasks, it will be helpful to know A2PO's performance on these tasks for better comparative evaluation.

---

> ### Author Response · Authors · 2023-11-22
> **Author Response (Part 1/3)**
>
> Thanks for the response!
>
>
> **[Comment 1]: CVAE increases the burden for hyperparameter tuning.**
> As a fact, in our A2PO implementation and experiments, the only hyperparameter of the advantage-aware CVAE is its training step $K$ with a hyperparameter search only over several values: $1\times10^5$, $2\times10^5$, $4\times10^5$, and $1\times10^6$. As shown in Table S4 of our paper, except for the specific requirements in the antmaze and maze2d environments, a value of $K=2*10^5$ proves to be sufficient for all other scenarios. This highlights the robustness of the CVAE training step across various tasks and datasets, without necessitating excessive hyperparameter tuning. Moreover, the number of training steps is a crucial hyperparameter for any deep learning model and is not unique to our CVAE.
>
> **[Comment 2]: the method does not offer impressive performance given its relatively heavy architecture and extra hyperparameters.**
>
> We appreciate your insights and are inclined to believe that an additional CVAE, in our view, is not a heavy architecture. Many previous lightweight methods like BCQ and BEAR, as well as recent state-of-the-art methods SPOT and LAPO, all utilize the CVAE to approximate the behavior policy. Our A2PO performance is superior to these baselines and comparable to the heavy diffusion-based methods. It is important to emphasize that the primary efficacy of A2PO stems from this advantage-aware policy, rather than the specific conditional model (be it CVAE or Diffusion). Specifically, we utilize $\pi(a|s,\xi)$ with the advantage value $\xi$ to represent the agent policy, while Diffusion-QL utilizes $\pi(a|s)$ to represent the agent policy. Diffusion-QL introduces a conditional diffusion with only the state as the condition to **implicitly** capture the multimodal action distribution, overlooking the intrinsic quality of different actions. However, our A2PO is conditioned on both the state and the advantage value. This advantage-aware policy **explicitly** leverages the advantage value to disentangle the action distributions of intertwined behavior policies, ensuring that the quality of different actions is positively correlated with the advantage value. Thus, our advantage-aware mechanism can also be effectively integrated with the diffusion-based methods, thereby enhancing their capacity to identify multi-modal data with different quality.
>
> **[Comment 3]: adding scores of these random-mixed datasets to the total score does not necessarily demonstrate the superiority of the proposed method in practice.**
>
> The motivation behind utilizing the combined datasets is to measure the ability of different offline baselines to learn from various types of multi-modal datasets, and more importantly, to reveal whether the agent is able to discriminate the high-quality data for effective learning and not be disturbed by the low-quality data. The final goal is only to highlight the effectiveness of our A2PO when there are substantial gaps between behavior policies. The reason for utilizing the random-incorporated datasets is that the datasets provided by D4RL have only a few types of structures, and we have to combine these finite datasets for more complicated and diverse datasets. Moreover, we give a summarization of the total scores in the locomotion tasks without the random-mixed datasets in Table R1 and Table R2. The results show that our A2PO achieves superior overall performance to the lightweight baselines. The performance of our A2PO is slightly worse than the diffusion-based method Diffusion-QL, but the benefits of diffusion mainly come from its heavyweight foundation model. As stated in comment 2 above, our approach is orthogonal to theirs; our core contribution is not the model itself, but the introduction of an advantage-aware policy.

---

> ### Author Response · Authors · 2023-11-22
> **Author Response (Part 2/3)**
>
> Table R1. Test returns of our proposed A2PO and lightweight baselines.
> | Env| BC| BCQ| TD3BC| CQL| IQL| MOPO| LAPO| XQL| SQL| EQL| A2PO|
> | ---------------------------- | ----------------- | ---------------- | ----------------- | ----------------- | ----------------- | ----------------- | ----------------- | ----------- | ----------- | ----------- | -------------------- |
> | halfcheetah-medium-v2        | 42.14 $\pm$ 0.33  | 46.83 $\pm$ 0.18 | 48.31 $\pm$ 0.10  | 47.20 $\pm$ 0.20  | 47.63 $\pm$ 0.05  | 48.40 $\pm$ 0.17  | 45.58 $\pm$ 0.06  | ***48.3***  | ***48.3***  | *47.2*      | 47.09 $\pm$ 0.17     |
> | hopper-medium-v2             | 50.45 $\pm$ 2.31  | 56.37 $\pm$ 2.74 | 58.55 $\pm$ 1.17  | 74.20 $\pm$ 0.82  | 51.17 $\pm$ 2.62  | 5.68 $\pm$ 4.00   | 52.53 $\pm$ 2.61  | *74.2*      | *75.5*      | *70.6*      | **80.29 $\pm$ 3.95** |
> | walker2d-medium-v2           | 71.73 $\pm$ 2.44  | 73.12 $\pm$ 1.38 | 83.62 $\pm$ 0.85  | 80.38 $\pm$ 0.77  | 63.75 $\pm$ 3.91  | 0.09 $\pm$ 0.06   | 80.46 $\pm$ 1.25  | *84.2*      | *84.2*      | *83.2*      | **84.88 $\pm$ 0.23** |
> | halfcheetah-medium-replay-v2 | 18.97 $\pm$ 13.85 | 40.87 $\pm$ 0.21 | 44.51 $\pm$ 0.22  | 46.74 $\pm$ 0.13  | 43.99 $\pm$ 0.33  | 37.46 $\pm$ 28.06 | 41.94 $\pm$ 0.47  | *45.2*      | ***44.8***  | *44.5*      | 44.74 $\pm$ 0.22     |
> | hopper-medium-replay-v2      | 20.99 $\pm$ 3.92  | 48.19 $\pm$ 5.52 | 65.20 $\pm$ 9.77  | 91.34 $\pm$ 1.99  | 52.61 $\pm$ 3.61  | 75.05 $\pm$ 28.82 | 50.14 $\pm$ 11.16 | *100.7*     | ***101.7*** | *98.1*      | 101.59 $\pm$ 1.25    |
> | walker2d-medium-replay-v2    | 13.99 $\pm$ 6.71  | 52.62 $\pm$ 4.62 | 81.28 $\pm$ 3.12  | 79.93 $\pm$ 1.26  | 68.84 $\pm$ 8.39  | 60.68 $\pm$ 19.32 | 60.55 $\pm$ 10.45 | *82.2*      | *77.2*      | *81.6*      | **82.82 $\pm$ 1.70** |
> | halfcheetah-medium-expert-v2 | 45.18 $\pm$ 1.22  | 46.87 $\pm$ 0.18 | 91.52 $\pm$ 1.82  | 16.47 $\pm$ 3.62  | 87.87 $\pm$ 1.97  | 69.73 $\pm$ 6.67  | 94.44 $\pm$ 0.46  | *94.2*      | *94.0*      | *94.6*      | **95.61 $\pm$ 0.54** |
> | hopper-medium-expert-v2      | 54.44 $\pm$ 4.05  | 58.05 $\pm$ 4.05 | 98.58 $\pm$ 2.48  | 89.19 $\pm$ 12.15 | 36.04 $\pm$ 21.36 | 20.32 $\pm$ 13.22 | 111.04 $\pm$ 0.36 | 111.2       | *110.8*| ***111.5*** | 105.44 $\pm$ 0.56    |
> | walker2d-medium-expert-v2    | 90.54 $\pm$ 5.93  | 75.14 $\pm$ 1.18 | 110.28 $\pm$ 0.26 | 102.65 $\pm$ 3.13 | 104.13 $\pm$ 0.76 | 91.92 $\pm$ 7.63  | 110.88 $\pm$ 0.15 | ***112.7*** | *111.0*| *110.2*     | 112.13 $\pm$ 0.24    |
> | **Total**| 408.43| 498.06| 681.85| 628.10| 556.03| 409.33| 647.56| 752.90| 747.50      | 741.50| **754.59**           |
>
>
> Table R2. Test returns of our proposed A2PO and diffusion-based baselines.
> | Env| Diffusion-QL | IDQL| A2PO|
> | ---------------------------- | ------------ | ----------- | --------------------- |
> | halfcheetah-medium-v2        | ***51.1***   | *51*        | 47.09 $\pm$ 0.17      |
> | hopper-medium-v2             | ***90.5***   | *65.4*      | 80.29 $\pm$ 3.95      |
> | walker2d-medium-v2           | ***87.0***   | *82.5*      | 84.88 $\pm$ 0.23      |
> | halfcheetah-medium-replay-v2 | ***47.8***   | *45.9*      | 44.74 $\pm$ 0.22      |
> | hopper-medium-replay-v2      | *95.5*       | *92.1*      | **101.59 $\pm$ 1.25** |
> | walker2d-medium-replay-v2    | ***101.3***  | *85.1*      | 82.82 $\pm$ 1.70      |
> | halfcheetah-medium-expert-v2 | ***96.8***   | *95.9*      | 95.61 $\pm$ 0.54      |
> | hopper-medium-expert-v2      | ***111.1***  | *108.6*     | 105.44 $\pm$ 0.56     |
> | walker2d-medium-expert-v2    | *110.1*      | ***112.7*** | 112.13 $\pm$ 0.24     |
> | Total| **791.20**   | 739.20      | 754.59|
>
>
> **[Comment 4]: it will be helpful to know A2PO's performance on antmaze-v0 or v2 tasks for better comparative evaluation.**
>
> We have evaluated A2PO on the antmaze-v0 tasks, as shown in Table R3. The results demonstrate that our A2PO achieves superior performance to other baselines in 'umaze' and 'medium' tasks.
>
> Table R3. Test returns of our proposed A2PO and other state-of-the-art baselines.
> | Env | BC   | BCQ  | BEAR | AWR  | AWAC | DT   | Onestep RL | TD3BC | CQL  | IQL  | PLAS | EQL  | Diffusion-QL | A2PO                 |
> | -------------------------- | ---- | ---- | ---- | ---- | ---- | ---- | ---------- | ----- | ---- | :--- | ---- | ---- | ------------ | -------------------- |
> | Antmaze-umaze-diverse-v0  | 45.6 | 55.0 | 61.0 | 70.3 | 49.3 | 53.0 | 60.7  | 71.4  | 84.0 | 62.2 | 45.3 | 82.0 | 66.2| **93.33 $\pm$ 9.43** |
> | Antmaze-medium-diverse-v0 | 0.0  | 0.0  | 8.0  | 0.0  | 0.0  | 0.0  | 0.0 | 3.0   | 53.7 | 70.0 | 0.7  | 73.6 | 78.6| **80.00 $\pm$ 8.16** |
> | Antmaze-large-diverse-v0  | 0.0  | 2.2  | 0.0  | 0.0  | 0.0  | 0.0  | 0.0  | 0.0   | 15.8 | 47.5 | 0.3  | 49.0 | **56.6**| 3.33 $\pm$ 4.71      |

---

> > ### Author Response · Authors · 2023-11-22
> > **Author Response (Part 3/3)**
> >
> > **We appreciate your time and effort in reviewing our work. We have taken your comments into careful consideration and conducted extensive experiments to address the concerns you raised. As suggested, we have additionally conducted comparison experiments on more recent SOTA methods. Moreover, we have also explained the reviewer-concerned implementation details as well as the stability of advantage-aware CVAE. Please let us know if you have other questions or comments.**
> >
> > **We sincerely look forward to your reevaluation of our work and would very appreciate it if you could raise your score to boost our chance of more exposure to the community. Thank you very much!**

---

> ### Author Response · Authors · 2023-11-23
> **Looking forward to your feedback**
>
> Dear Reviewer,
>
> We deeply appreciate the time and effort you have invested in reviewing our manuscript. Your expertise and insights are invaluable to us.
>
> As the discussion period is nearing its conclusion, with less than **two hours** remaining, we are keen to know if our responses have adequately addressed your concerns.
>
> Thank you once again for your attention and consideration. We eagerly await your response.
>
> Sincerely,
>
> Authors of A2PO

---

### Official Review · Reviewer_Mmfe · 2023-10-28

**Soundness:** 2 fair
**Presentation:** 3 good
**Contribution:** 2 fair
**Rating:** 5
**Confidence:** 4

**Summary:**

This manuscript proposes an Advantage-Aware Policy Optimization (A2PO) method for offline RL problem. The authors first disentangled behavior policies with CVAE, then combine the  advantage-aware method for policy improvement and optimization.  The superiority of the A2PO method is validated from single-quality and mixed-quality datasets of the D4RL.

**Strengths:**

1. The problem of mixed behavior policy data is very interesting, and the main idea of this manuscript is easy to follow.
2. The empirical studies shows that the A2PO method significantly outperforms the competitors in most cases.

**Weaknesses:**

1. Some details in methodologies are not clear.  How to select the $\xi^*$ in equation (7) and the optimal selected action $\alpha^*_\xi$. As the $\xi$ is transformed by the tanh function, what if there are no (s,a) pairs with $\xi^*=1$? This is an very important step for understanding the main idea of this paper.
2. Follow the last point, if the $\xi$ is normalized to [0,1], the largest value of $\xi=1$ only corresponds to single data point in the mixed dataset, how do you learn the action distribution from CVAE with this single data point?
3. The information in pseudocode in Algorithm 1 is limited, it is suggested to add some important details in Algorithm.

**Questions:**

1. Some offline-RL baselines, such as BEAR/AWAC are missed in experiment, and the recent popular SPOT [1] method is not considered in experiments as well.  It is suggested to consider the baselines in offline-RL methods.
2. The author suggested CVAE to disentangle policies from mixed data. However, the CVAE may not be a good choice for distinguish data from different sources from [1] and [2]. So I wonder why A2PO could achieve significant results in empirical experiments.
3. In the experiment, the authors manually combines the single-quality datasets to setup the mixed datasets. The details of mixed data combination should be provided, and some ablation studies about the amount of each single-quality data are also suggested.

[1] Supported Policy Optimization for Offline Reinforcement Learning. https://arxiv.org/abs/2202.06239

[2] A Behavior Regularized Implicit Policy for Offline Reinforcement Learning. https://arxiv.org/pdf/2202.09673.pdf

[3] EMaQ: Expected-Max Q-Learning Operator for Simple Yet Effective Offline and Online RL. https://arxiv.org/pdf/2007.11091.pdf

---

> ### Author Response · Authors · 2023-11-19
> **Author Response (Part 1/3)**
>
> We greatly appreciate the reviewer for the constructive comments, which help us improve the quality of the paper significantly. Below we address the main points raised in the review. We will update the additional experimental results in the revision later.
>
> **[Weakness 1 \& Weakness 2]:  more details about the largest advantage condition $\xi^\*$**
>
> Sorry for the confusion. Our advantage-aware CVAE disentangles and infers the action distributions of intertwined behavior policies by modeling the advantage values of all training data as conditional variables.
>
> Concretely, Unlike previous methods [1] utilizing CVAE to predict action solely based on the state $s$, we consider both state $s$ and advantage value $\xi$ for the CVAE condition.  For a state-action pair $(s, a)$, the advantage condition $\xi$ can be computed with agent critic as $\xi =\tanh(\min\_{i=1,2} Q\_{\theta\_{i}}(s,a)- V\_\phi(s))$. The state-advantage condition $c$ is formulated as $c=s\parallel\xi$, while the CVAE model is trained using the state-advantage condition $c$ and the corresponding action $a$ with ELBO loss function. Since the range $\xi$ is normalized within $(-1,1)$ with $\tanh$, with the fixed largest advantage condition $\xi^*=1$ the CVAE is able to infer high-quality action distributions associated with the behavior policies from the CVAE decoder. As for the advantage-aware policy $\pi\_\omega(\cdot|c)$, it generates a latent representation $\tilde{z}$ based on the state-advantage condition $c$.  $\tilde{z}$ is then fed into the CVAE decoder to get a recognizable action. The optimal action $a^*\_\xi$ is obtained with the designated $\xi^*=1$ condition and the corresponding latent representation $\tilde z^*$ from the policy network.
>
> In short, our CVAE directly learns from all training data with different advantage values, benefiting from the generalization capability to infer the high-advantage action distribution. Considering Equation (7), the action $a^*\_\xi$ with designated $\xi^*=1$ is not selected directly from the dataset samples. Instead, the CVAE decoder infers the high-advantage action distribution only based on the state $s$ and fixed advantage value $\xi^*$. Then the agent can select the optimal action $a^*$ from this distribution.
>
>
>
> **[Weakness 3]: It is suggested to add some important details in Algorithm 1.**
>
> Thanks for the suggestion. We have added details in Algorithm 1, illustrating how $c$ and $c^*$ are computed for both behavior policy disentangling and agent policy optimization.

---

> ### Author Response · Authors · 2023-11-19
> **Author Response (Part 2/3)**
>
> **[Question 1]: It is suggested to consider the baselines in offline-RL methods.**
>
> As suggested by the reviewer, we have added the mentioned baselines for comparison, including BEAR, AWAC, and SPOT, as shown in Table R1. The results show that our A2PO exhibits superior performances compared to the baselines, especially in the mixed-quality datasets.
>
> Table R1. Test returns of our proposed A2PO and the suggested baselines on the locomotion tasks. The *italicized* results of BEAR are obtained from the D4RL paper [3], while the *italicized* results of AWAC and SPOT are obtained from the SPOT paper [1]. The other results are evaluated over 3 random seeds based on the corresponding official code.
>
> | Env                                 | BEAR             | AWAC                 | SPOT                | A2PO                  |
> | ----------------------------------- | ---------------- | -------------------- | ------------------- | --------------------- |
> | halfcheetah-medium-v2               | *41.7*           | *43.5*               | ***58.4***          | 47.09 $\pm$ 0.17      |
> | hopper-medium-v2                    | *52.1*           | *57.0*               | ***86.0***          | 80.29 $\pm$ 3.95      |
> | walker2d-medium-v2                  | *59.1*           | *72.4*               | ***86.4***          | 84.88 $\pm$ 0.23      |
> | halfcheetah-medium-replay-v2        | *38.6*           | *40.5*               | ***52.2***          | 44.74 $\pm$ 0.22      |
> | hopper-medium-replay-v2             | *19.2*           | *37.2*               | *100.2*             | **101.59 $\pm$ 1.25** |
> | walker2d-medium-replay-v2           | *33.7*           | *27.0*               | ***91.6***          | 82.82 $\pm$ 1.70      |
> | halfcheetah-medium-expert-v2        | *53.4*           | *42.8*               | *86.9*              | **95.61 $\pm$ 0.54**  |
> | hopper-medium-expert-v2             | *40.1*           | *55.8*               | *99.3*              | **105.44 $\pm$ 0.56** |
> | walker2d-medium-expert-v2           | *96.3*           | *74.5*               | *112.0*             | **112.13 $\pm$ 0.24** |
> | halfcheetah-random-expert-v2        | 4.18 $\pm$ 1.79  | 87.32 $\pm$ 2.91     | 60.42 $\pm$ 3.69    | **90.32 $\pm$ 1.63**  |
> | hopper-random-expert-v2             | 5.73 $\pm$ 3.63  | 84.70 $\pm$ 19.83    | 98.60 $\pm$ 5.32    | **105.19 $\pm$ 4.54** |
> | walker2d-random-expert-v2           | -0.36 $\pm$ 0.00 | 11.70 $\pm$ 12.04    | 7.20 $\pm$ 9.96     | **91.96 $\pm$ 10.98** |
> | halfcheetah-random-medium-v2        | 13.78 $\pm$ 4.87 | **46.54 $\pm$ 0.10** | 46.12 $\pm$ 0.25    | 45.20 $\pm$ 0.21      |
> | hopper-random-medium-v2    | 1.40 $\pm$ 0.58  | 19.45 $\pm$ 11.87    | **7.78 $\pm$ 0.87** | 7.14 $\pm$ 0.35       |
> | walker2d-random-medium-v2       | -0.46 $\pm$ 0.07 | -0.03 $\pm$ 0.08     | 7.77 $\pm$ 4.10     | **75.80 $\pm$ 2.12**  |
> | halfcheetah-random-medium-expert-v2 | 2.25 $\pm$ 0.00  | 89.65 $\pm$ 1.67     | 80.18 $\pm$ 7.47    | **90.58 $\pm$ 1.44**  |
> | hopper-random-medium-expert-v2      | 8.62 $\pm$ 2.10  | 30.31 $\pm$ 7.46     | 40.19 $\pm$ 23.07   | **107.84 $\pm$ 0.42** |
> | walker2d-random-medium-expert-v2    | -0.41 $\pm$ 0.65 | -0.31 $\pm$ 0.07     | 10.31 $\pm$ 4.63    | **97.71 $\pm$ 6.74**  |
> | **Total** | 468.93 | 1205.23 | 1131.57   | **1466.33**  |
>
>
>
> **[Question 2]: The author suggested CVAE to disentangle policies from mixed data. However, the CVAE may not be a good choice for distinguish data from different sources from [1] and [2]. So I wonder why A2PO could achieve significant results in empirical experiments.**
>
> Thanks for the insightful comments. Our A2PO can utilize CVAE to disentangle behavior policies from mixed data for achieving superior performance, primarily due to the distinct conditional variables of our CVAE in comparison to those described in the CVAE of reference [1, 2].
>
> (1) Firstly, it should be noted that the state-of-the-art SPOT [1] does not illustrate the limitations of CVAE in distinguishing data. Instead, SPOT also utilizes CVAE to estimate behavior policy outputs, which are then used to provide the constraint penalty term. Reference [2] employs a behavior cloning experiment to evaluate the performance of the CVAE, which reveals that the CVAE overestimates unwanted low data-density regions. However, **the CVAE of reference [1, 2] conditiones only on the given state**, which tries to distinguish data from different sources without discrimination of the data quality.
>
> (2) In contrast, our advantage-aware CVAE disentangles different behavior policies into separate action distributions by **modeling the advantage values of collected state-action pairs as conditioned variable**. By conditioning on specific states and advantage values, our CVAE generates corresponding actions, ensuring that the quality of these actions is positively correlated with the advantage value. Thus, A2PO can further optimize the agent toward high advantage values to obtain an effective decision-making policy.

---

> ### Author Response · Authors · 2023-11-19
> **Author Response (Part 3/3)**
>
> **[Question 3]: The details of mixed data combination should be provided, and some ablation studies about the amount of each single-quality data are also suggested.**
>
> Sorry for the confusion. As stated in Section 5.1 (Tasks and Datasets) of our original paper, we construct mixed datasets using single-quality datasets from the D4RL benchmark. The single-quality datasets are generated with random, medium, and expert behavior policies. Since the D4RL benchmark only includes the two mixed-quality datasets ("medium-replay" and "medium-expert"), we follow the official instructions provided by D4RL (https://github.com/Farama-Foundation/D4RL/wiki/Dataset-Reproducibility-Guide) to manually construct the other mixed-quality datasets ("random-medium", "random-expert", and "random-medium-expert") by directly combining the corresponding single-quality datasets. Each single-quality dataset ("random", "medium", "expert") has 1e6 transitions. The mixed-quality "random-medium" and "random-expert" datasets consist of 2e6 transitions, while the "random-medium-expert" dataset contains 3e6 transitions.
>
> As suggested by the reviewer, we have additionally conducted an ablation study to investigate the impact of varying amounts of single-quality samples in mixed-quality datasets, as shown in Table R2. The results demonstrate the robustness of our A2PO model in handling mixed-quality datasets containing different proportions of single-quality samples.
>
> Table R2. Test returns of our proposed A2PO on the locomotion tasks with the medium-expert dataset containing different proportions of single-quality samples.
> |  Env           | medium:expert=3:1 | medium:expert=2:1 | medium:expert=1:1     | medium:expert=1:2     | medium:expert=1:3 |
> | ----------- | ----------------- | ----------------- | --------------------- | --------------------- | ----------------- |
> | halfcheetah | 93.78 $\pm$ 0.70  | 94.28 $\pm$ 0.09  | **95.61 $\pm$ 0.54**  | 95.30 $\pm$ 0.25      | 95.06 $\pm$ 0.22  |
> | hopper      | 106.94 $\pm$ 0.67 | 74.72 $\pm$ 25.18 | 105.44 $\pm$ 0.56     | **112.19 $\pm$ 0.09** | 110.17 $\pm$ 1.37 |
> | walker2d    | 110.94 $\pm$ 0.36 | 110.77 $\pm$ 0.57 | **112.13 $\pm$ 0.24** | 111.26 $\pm$ 0.49     | 110.67 $\pm$ 0.21 |
> | **Total**                               | 311.66           | 279.77     | 313.18  | **318.75**    | 315.90 |
>
> **References**
>
> [1] Wu, J. et al. Supported Policy Optimization for Offline Reinforcement Learning. NeurIPS 2022.
>
> [2] Yang, S. et al. A Behavior Regularized Implicit Policy for Offline Reinforcement Learning. arXiv 2022.
>
> [3] Fu, J. et al. D4RL: Datasets for Deep Data-Driven Reinforcement Learning. arXiv 2020.

---

> > ### Author Response · Authors · 2023-11-22
> > **Auther Response**
> >
> > Dear reviewer,
> >
> > We are glad that the reviewer appreciates our attempt, and sincerely thank the reviewer for the constructive comments. As suggested, we have additionally conducted comparison experiments on more state-of-the-art baselines with more diverse datasets. Moreover, we have also explained the details about the advantage condition and the effectiveness of our advantage-aware CVAE. Please let us know if you have other questions or comments.
> >
> > Since the discussion window between reviewers and authors is ending, we sincerely look forward to your reevaluation of our work and would very appreciate it if you could raise your score to boost our chance of more exposure to the community. Thank you very much!
> >
> > Best regards,
> >
> > The authors of A2PO

---

> ### Author Response · Authors · 2023-11-23
> **Looking forward to your feedback**
>
> Dear Reviewer,
>
> We deeply appreciate the time and effort you have invested in reviewing our manuscript. Your expertise and insights are invaluable to us.
>
> As the discussion period is nearing its conclusion, with less than **two hours** remaining, we are keen to know if our responses have adequately addressed your concerns.
>
> Thank you once again for your attention and consideration. We eagerly await your response.
>
> Sincerely,
>
> Authors of A2PO

---

### Official Review · Reviewer_R6oW · 2023-10-30

**Soundness:** 2 fair
**Presentation:** 2 fair
**Contribution:** 2 fair
**Rating:** 5
**Confidence:** 4

**Summary:**

This paper presents an offline advantage-aware policy optimization method, which uses a generative model to describe the action distribution conditioned on states and advantage values, then samples from this model for high-advantage actions to conduct offline policy optimization. The method is motivated by the argument that mixed-quality dataset may have conflicting constraints if each sample is treated equally, hence some differentiation is needed. The proposed advantage-aware differentiation is verified on datasets of single- and mixed-quality. The empirical results show their method outperforms baselines and especially works well on a dataset with mixed-quality. Ablations are also provided to analyze the method design choices.

**Strengths:**

1. The empirical results are strong compared to baselines.
2. The paper presents a detailed ablation analysis for the proposed method.

**Weaknesses:**

1. The contribution is a bit incremental, by mainly moving the advantage values from weight coefficient of the loss function (as in LAPO equation [7]) to condition variables (as in this paper equation [5]).
2. It looks to me that the policy optimization part is not the same as LAPO, which uses TD3. The differences make the performance comparison between A2PO and LAPO not fair, and make it hard to justify the benefits of conditioning the generative model using advantage values.

**Questions:**

1. A minor typo on Page 8: “The performance comparison of different discrete advantage conditions for test is given in Figure **4**”.
2. I wonder how A2PO compares to LAPO if the policy optimization steps are made the same.

---

> ### Author Response · Authors · 2023-11-19
> **Author Response (Part 1/2)**
>
> We sincerely thank the reviewer for the constructive comments, which help us improve the quality of the paper significantly. Below we address the main points raised in the review. We will update the additional experimental results in the revision later.
>
>
> **[Weakness 1]: the contribution about the advantage values**
>
> Sorry for the confusion. Our main contribution is the first dedicated attempt toward advantage-aware policy optimization to alleviate the constraint conflict issue under the mixed-quality offline dataset. LAPO [1] and our A2PO solve the constraint conflict issue from two quite different aspects.
>
> (1) Methodology. LAPO is an advantage-weighted method, which employs weighted sampling to prioritize training transitions with high advantage values from the offline dataset. In contrast, our proposed A2PO is an advantage-aware method, which directly conditions the agent policy on the advantage values of all training data without any prior preference.
>
> (2) Effectiveness. The advantage-weighted mechanism of LAPO implicitly reduces the diverse behavior policies associated with the offline dataset into a narrow one from the viewpoint of the dataset redistribution. LAPO inherently biases the learning process towards transitions with higher advantage values, potentially overlooking critical information from lower-advantage transitions. As a result, this redistribution operation can be overfitting on high-advantage data during training, thus impeding the advantage estimation for the effective state-action space.  These errors in advantage estimation can further lead to unreliable policy optimization. In contrast, our proposed A2PO disentangles different behavior policies into separate action distributions by modeling the advantage values of collected state-action pairs as conditioned variables, which maintains an unbiased perspective of the entire dataset. Thus, A2PO can perform a more effective advantage estimation than LAPO, as illustrated in Figure 1 and Figure S4 of our original paper. Furthermore,  A2PO exhibits superior performances compared to LAPO, as illustrated in Table 1 of our original paper.
>
> In summary, "moving the advantage values from the weight coefficient of the loss function to condition variables" represents a significant conceptual leap, not merely a positional change. While LAPO and A2PO share the common goal of addressing the constraint conflict in mixed-quality datasets, the methodology and effectiveness of our A2PO are distinct and innovative, offering a novel perspective on advantage-aware policy optimization.

---

> ### Author Response · Authors · 2023-11-19
> **Author Response (Part 2/2)**
>
> **[Weakness 2 \& Question 2]: the differences in policy optimization between A2PO and LAPO**
>
> Thanks for the constructive comments. We would like to point out that A2PO and LAPO stand for advantage-aware and advantage-weighted methods, respectively. These methods do not necessarily require identical structures or loss functions in policy optimization.
>
> Technically, LAPO only adopts the deterministic policy gradient loss in policy optimization, while our A2PO adopts the TD3BC-style loss based on advantage conditions, including the deterministic policy gradient loss and the behavior cloning loss. The first term encourages the optimal policy condition on the largest advantage condition $\xi^*$ to select actions that yield the highest expected returns represented by the Q-value. The second behavior cloning term explicitly imposes constraints on the advantage-aware policy, ensuring the policy selects in-sample actions that adhere to the advantage condition $\xi$ determined by the critic.
>
>
> We have additionally conducted experiments of A2PO with and without the BC term in the policy optimization step, as shown in Table R1. The results indicate that even without the BC regularization term, A2PO consistently outperforms LAPO in the majority of tasks. Moreover, the BC term in A2PO can enhance its performance in most cases. This comparison highlights the superior performance achieved by A2PO, showcasing its effective disentanglement of action distributions from different behavior policies in order to enforce a reasonable advantage-aware policy constraint and obtain an optimal agent policy.
>
>
> Table R1. Test returns of LAPO and our proposed A2PO without BC term in policy optimization step in the locomotion tasks. The results of A2PO w/o BC are evaluated over 3 random seeds while the LAPO and A2PO results in our original paper are evaluated over 5 random seeds. **Bold** indicates that the better performance among LAPO and A2PO w/o BC.
> |  Environment                                 | LAPO                  | A2PO w/o BC           | A2PO              |
> | ----------------------------------- | --------------------- | --------------------- | ----------------- |
> | halfcheetah-medium-v2               | 45.58 $\pm$ 0.06      | **46.81 $\pm$ 0.09**  | 47.09 $\pm$ 0.17  |
> | hopper-medium-v2                    | 52.53 $\pm$ 2.61      | **70.08 $\pm$ 4.03**  | 80.29 $\pm$ 3.95  |
> | walker2d-medium-v2                  | 80.46 $\pm$ 1.25      | **81.97 $\pm$ 1.12**  | 84.88 $\pm$ 0.23  |
> | halfcheetah-medium-replay-v2        | 41.94 $\pm$ 0.47      | **42.01 $\pm$ 0.28**  | 44.74 $\pm$ 0.22  |
> | hopper-medium-replay-v2             | 50.14 $\pm$ 11.16     | **96.51 $\pm$ 1.47**  | 101.59 $\pm$ 1.25 |
> | walker2d-medium-replay-v2           | 60.55 $\pm$ 10.45     | **71.09 $\pm$ 7.98**  | 82.82 $\pm$ 1.70  |
> | halfcheetah-medium-expert-v2        | 94.22 $\pm$ 0 .46     | **94.29 $\pm$ 0.03**  | 95.61 $\pm$ 0.54  |
> | hopper-medium-expert-v2             | **111.04 $\pm$ 0.36** | 107.27 $\pm$ 1.93     | 105.44 $\pm$ 0.56 |
> | walker2d-medium-expert-v2           | 110.88 $\pm$ 0.15     | **111.61 $\pm$ 0.07** | 112.13 $\pm$ 0.24 |
> | halfcheetah-random-expert-v2        | **52.58 $\pm$ 17.30** | 31.37 $\pm$ 6.27      | 90.32 $\pm$ 1.63  |
> | hopper-random-expert-v2             | 82.33 $\pm$ 18.95     | **113.20 $\pm$ 1.20** | 105.19 $\pm$ 4.54 |
> | walker2d-random-expert-v2           | 0.89 $\pm$ 0.53       | **66.82 $\pm$ 11.01** | 91.96 $\pm$ 10.98 |
> | halfcheetah-random-medium-v2        | 18.53 $\pm$ 0.99      | **43.19 $\pm$ 0.54**  | 45.20 $\pm$ 0.21  |
> | hopper-random-medium-v2             | **4.17 $\pm$ 3.11**   | 1.59 $\pm$ 0.92       | 7.14 $\pm$ 0.35   |
> | walker2d-random-medium-v2           | 23.65 $\pm$ 33.97     | **72.32 $\pm$ 4.43**  | 75.80 $\pm$ 2.12  |
> | halfcheetah-random-medium-expert-v2 | **71.09 $\pm$ 0.47**  | 70.77 $\pm$ 4.21      | 90.58 $\pm$ 1.44  |
> | hopper-random-medium-expert-v2      | 66.59 $\pm$ 19.29     | **86.54 $\pm$ 7.25**  | 107.84 $\pm$ 0.42 |
> | walker2d-random-medium-expert-v2    | 60.41 $\pm$ 43.22     | **110.35 $\pm$ 1.20** | 97.71 $\pm$ 6.74  |
> | **Total**  | 1027.58   | **1317.79** | 1466.33  |
>
>
>
> **[Question 1]: a minor typo**
>
> Thanks. We have corrected it in the paper.
>
>
> **References**
>
> [1] Chen, X. et al. LAPO: Latent-Variable Advantage-Weighted Policy Optimization for Offline Reinforcement Learning. NeurIPS 2022.

---

> > ### Author Response · Authors · 2023-11-22
> > **Author Response**
> >
> > Dear reviewer,
> >
> > We are glad that the reviewer appreciates our attempt, and sincerely thank the reviewer for the constructive comments. As suggested, we have additionally conducted experiments on A2PO without the BC regularization term. Moreover, we have also explained our contribution of advantage-aware policy as well as illustrated the policy optimization difference between our A2PO and the advantage-weighted method LAPO. Please let us know if you have other questions or comments.
> >
> > Since the discussion window between reviewers and authors is ending, we sincerely look forward to your reevaluation of our work and would very appreciate it if you could raise your score to boost our chance of more exposure to the community. Thank you very much!
> >
> > Best regards,
> >
> > The authors of A2PO

---

> ### Author Response · Authors · 2023-11-23
> **Looking forward to your feedback**
>
> Dear Reviewer,
>
> We deeply appreciate the time and effort you have invested in reviewing our manuscript. Your expertise and insights are invaluable to us.
>
> As the discussion period is nearing its conclusion, with less than **two hours** remaining, we are keen to know if our responses have adequately addressed your concerns.
>
> Thank you once again for your attention and consideration. We eagerly await your response.
>
> Sincerely,
>
> Authors of A2PO

---

### Official Review · Reviewer_TZv3 · 2023-10-30

**Soundness:** 3 good
**Presentation:** 4 excellent
**Contribution:** 3 good
**Rating:** 6
**Confidence:** 3

**Summary:**

This paper develops an offline RL method for mixed-quality datasets. Their algorithm alternates between (1) performing actor-critic-style policy optimization and (2) disentangling policy quality within the latent space of the actor. (2) is accomplished by CVAE training in which prediction is conditioned on both the state and the current advantage value. The novelty of this paper is the choice to condition on the current advantage value. They show that their method (A2PO) outperforms baselines such as CQL, IDL, MOPO, LAPO, etc on D4RL navigation and locomotion tasks.

**Strengths:**

This is a sound and well-presented paper. I would place the readability and cleanliness of presentation in the top 5% of papers. A2PO outperforms baselines on most D4RL navigation and locomotion tasks. The ablation analysis in Figure 4 shows that the resulting policy is very responsive to advantage conditioning. I think this is a powerful and elegant way of handling the underestimation issue that has been recurrent in work on off-line RL. I think the experimental results and algorithmic contribution will be useful to the offline RL community.

**Weaknesses:**

The main weakness of this paper is lack of empirical depth. Many of the baselines also show strong performance on sparse-reward/manipulation tasks within D4RL and it's not clear why these tasks have been omitted here. It would be helpful for the authors to clarify if this approach is specific to navigation and locomotion or to show results on a manipulation tasks such as FrankaKitchen or Adroit. For a concrete reference, please see Table 2 of the CQL paper.

The idea of conditioning on a proxy for policy return (i.e., the advantage) itself is not novel. It would be helpful for the authors to include a reference to reward-conditioned supervised learning [Brandfonbrener et. al., 2023].

- Brandfonbrener et. al. When does return-conditioned supervised learning work for offline reinforcement learning? https://arxiv.org/abs/2206.01079

**Questions:**

Why do the authors ablate discrete vs continuous advantage variables within the CVAE? For a distribution that is inherently categorical, I could understand why this ablation would be useful, but because the space of advantages is continuous anyways I don't understand what the reader should takeaway from this experiment.

The margins for improvement between A2PO and the baselines is quite narrow in some cases. It's possible that the ordering of the results could change with more or less tuning. Can the authors clarify what hyper parameters were tuned within A2PO vs the baselines?

---

> ### Author Response · Authors · 2023-11-19
> **Author Response (Part 1/2)**
>
> We are glad that the reviewer appreciates our work as a useful contribution to the community. Below we address the main points raised in the review. We will update the additional experimental results in the revision later.
>
>
>
> **[Weakness 1]: more experiments in the robot manipulation environments**
>
> Thanks for the constructive suggestions. We have additionally conducted experiments on the robot manipulation tasks of Adroit and FrankaKitchen, as shown in Table R1. Besides the baselines presented in our original paper, the recent state-of-the-art baseline SPOT [1] is also included for a more comprehensive comparison. The results demonstrate that our A2PO method can achieve comparable performance to the state-of-the-art LAPO [2] and SPOT [1], as well as the other baselines. This underscores the versatility of our A2PO method, highlighting its applicability not only in navigation and locomotion tasks but also in a broader spectrum of robotic manipulation tasks.
>
>
> Table R1. Test returns of our proposed A2PO and baselines in the robot manipulation tasks of Adroit and FrankaKitchen. *Italics* indicates that the results are obtained from the D4RL paper [3], while the other results are evaluated over 3 random seeds based on the corresponding official code.
> | Environment         | BC     | BCQ    | CQL       | IQL        | TD3BC            | SPOT              | BPPO              | LAPO                  | A2PO                  |
> | ------------------- | ------ | ------ | --------- | ---------- | ---------------- | ----------------- | ----------------- | --------------------- | --------------------- |
> | kitchen-complete-v0 | *33.8* | *8.1*  | *43.8*    | ***62.5*** | 0.83 $\pm$ 1.18  | 41.67 $\pm$ 11.40 | 0.00 $\pm$ 0.00   | 50.83 $\pm$ 10.27     | 60.00 $\pm$ 2.04      |
> | kitchen-partial-v0  | *33.8* | *18.9* | *49.8*    | *46.3*     | 0.00 $\pm$ 0.00  | 0.00 $\pm$ 0.00   | 16.67 $\pm$ 2.36  | **58.75 $\pm$ 16.25** | 48.33 $\pm$ 4.08      |
> | kitchen-mixed-v0    | *47.5* | *10.6* | *51.0*    | *51.0*     | 0.83 $\pm$ 1.18  | 0.00 $\pm$ 0.00   | 0.00 $\pm$ 0.00   | 52.33 $\pm$ 6.27      | **53.33 $\pm$ 2.36**  |
> | pen-human-v1        | *34.4* | *12.3* | *37.5*    | ***71.5*** | -3.69 $\pm$ 0.38 | 32.70 $\pm$ 11.40 | 25.77 $\pm$ 19.25 | 68.06 $\pm$ 18.01     | 68.94 $\pm$ 5.90      |
> | hammer-human-v1     | *1.5*  | *1.2*  | ***4.4*** | *1.4*      | 0.71 $\pm$ 0.25  | 1.99 $\pm$ 0.07   | 0.50 $\pm$ 0.35   | 1.12 $\pm$ 0.29       | 1.84 $\pm$ 0.35       |
> | door-human-v1       | *0.5*  | *0.4*  | ***9.9*** | *4.3*      | -0.33 $\pm$ 0.01 | -0.33 $\pm$ 0.01  | -0.02 $\pm$ 0.03  | 6.07 $\pm$ 4.60       | 8.51 $\pm$ 3.66       |
> | relocate-human-v1   | *0.0*  | *0.0*  | *0.2*     | *0.1*      | -0.30 $\pm$ 0.01 | -0.07 $\pm$ 0.13  | -0.08 $\pm$ 0.08  | 0.04 $\pm$ 0.01       | **0.49 $\pm$ 0.58**   |
> | pen-cloned-v1       | *56.9* | *28.0* | *39.2*    | *37.3*     | 1.74 $\pm$ 2.90  | 2.54 $\pm$ 7.96   | 21.77 $\pm$ 5.88  | 55.84 $\pm$ 20.29     | **84.80 $\pm$ 21.43** |
> | hammer-cloned-v1    | *0.8*  | *0.4*  | ***2.1*** | ***2.1***  | 0.26 $\pm$ 0.02  | 0.46 $\pm$ 0.03   | 0.25 $\pm$ 0.22   | 0.84 $\pm$ 0.35       | 0.43 $\pm$ 0.15       |
> | door-cloned-v1      | *-0.1* | *0.0*  | *0.4*     | ***1.6***  | -0.35 $\pm$ 0.02 | -0.35 $\pm$ 0.02  | -0.04 $\pm$ 0.05  | 0.22 $\pm$ 0.40       | 0.31 $\pm$ 0.04       |
> | relocate-cloned-v1  | -0.1   | *-0.2* | -0.1      | *-0.2*     | -0.31 $\pm$ 0.01 | -0.30 $\pm$ 0.01  | -0.14 $\pm$ 0.11  | -0.05 $\pm$ 0.07      | **0.00 $\pm$ 0.04**   |
> | **Total**  | 209.00   | 79.70 | 238.20  | 277.90   | -0.61 | 78.31  | 64.68  | 294.05  | **326.98**  |
>
>
>
> **[Weakness 2]: more discussion about reward-conditioned supervised learning**
>
> Thanks for the insightful suggestions. As suggested by the reviewer, we have discussed the paper "When does return-conditioned supervised learning work for offline reinforcement learning?" [4] in the related works. The return-conditioned method relies on sampling entire trajectories to obtain return signals for supervised learning. In contrast, our A2PO method can leverage arbitrary transition samples to estimate the normalized advantage value for CVAE training. This eliminates the need for explicit trajectory return signals, rendering our approach more adaptable and practical for real-world applications where only transition data is accessible and the returns of entire trajectories are not known.

---

> ### Author Response · Authors · 2023-11-19
> **Author Response (Part 2/2)**
>
> **[Question 1]: Why do the authors ablate discrete vs continuous advantage variables within the CVAE?**
>
> Sorry for the confusion. The ablation analysis in Section 5.3 of the original paper demonstrates the effectiveness of the continuous advantage condition with the normalization trick, which enables the CVAE to disentangle mixed-quality datasets. Moreover, the discrete advantage condition is proposed to stabilize the normalized advantage value, as the advantage value of a given state-action pair may fluctuate during different training phases. Nonetheless, the ablation study indicates that such discretization can only enhance agent performance in several simple tasks. This can be attributed to the fact that discretization potentially impedes the accurate estimation of the advantage values, especially in complex tasks.
>
>
> **[Question 2]: Can the authors clarify what hyper parameters were tuned within A2PO vs the baselines?**
>
>
> Sorry for the confusion. We only tune two hyperparameters in our experiments: the use of discrete $\xi$ and the CVAE training step $K$, as indicated in Table S4 in the original paper. For discrete $\xi$, we observed that the simple tasks, specifically antmaze and maze2d, can benefit from its utilization. In contrast, for the other tasks, the direct use of continuous $\xi$ yielded state-of-the-art results. For the CVAE training step $K$, we conducted a hyperparameter search over $1\times10^5$, $2\times10^5$, $4\times10^5$, and $1\times10^6$. Except for the specific requirements in the antmaze and maze2d environments, a value of $K=2*10^5$ proves to be sufficient for all other scenarios. In short, these two hyperparameters showcase robustness across different tasks and datasets, requiring only minimal adjustments. The remaining hyperparameter settings of A2PO follow the official TD3BC [5] implementation for actor-critic and the official LAPO [2] implementation for CVAE.
>
>
> **References**
>
> [1] Wu, J. et al. Supported Policy Optimization for Offline Reinforcement Learning. NeurIPS 2022.
>
> [2] Chen, X. et al. LAPO: Latent-Variable Advantage-Weighted Policy Optimization for Offline Reinforcement Learning. NeurIPS 2022.
>
> [3] Fu, J. et al. D4RL: Datasets for Deep Data-Driven Reinforcement Learning. arXiv 2020.
>
> [4] Brandfonbrener, D. et al. When does return-conditioned supervised learning work for offline reinforcement learning? NeurIPS 2022.
>
> [5] Fujimoto, S., & Gu, S. S. A minimalist approach to offline reinforcement learning. NeurIPS 2021.

---

> > ### Author Response · Authors · 2023-11-22
> > **Author Response**
> >
> > Dear reviewer,
> >
> > We are glad that the reviewer appreciates our attempt, and sincerely thank the reviewer for the constructive comments. As suggested, we have additionally conducted comparison experiments on more robot tasks. Moreover, we have also explained the relation to the reward-conditioned supervised learning, the motivation of the ablation study about the advantage signal, as well as our hyperparameters setting. Please let us know if you have other questions or comments.
> >
> > Since the discussion window between reviewers and authors is ending, we sincerely look forward to your reevaluation of our work and would very appreciate it if you could raise your score to boost our chance of more exposure to the community. Thank you very much!
> >
> > Best regards,
> >
> > The authors of A2PO

---

> > > ### Comment · Reviewer_TZv3 · 2023-11-22
> > >
> > > Hi, Thank you for your considerate response. I feel my concerns were well addressed and still lean towards accepting this paper. I hesitate to raise my score to "Accept" from "Weak Accept" because I am curious to hear from reviewer HeZ6 if they feel the new baselines added adequately address their concerns. Otherwise I feel the issues raised have been well addressed.

---

> > > > ### Author Response · Authors · 2023-11-23
> > > > **Thanks for the positive support!**
> > > >
> > > > Dear Reviewer,
> > > >
> > > > We sincerely appreciate your positive feedback, the time and effort you have put into reviewing our modifications and responses! It is immensely gratifying to learn that our revisions have successfully addressed your concerns. We understand your hesitation in raising the score according to further comments from reviewer HeZ6 regarding the new baselines. We have made every effort to comprehensively address the concerns raised by HeZ6.
> > > >
> > > > Thank you once again for your constructive feedback and support.
> > > >
> > > > Best,
> > > >
> > > > Authors

---

### Official Review · Reviewer_KGRx · 2023-11-01

**Soundness:** 3 good
**Presentation:** 3 good
**Contribution:** 2 fair
**Rating:** 6
**Confidence:** 4

**Summary:**

The authors introduce A2PO, which aims to learn an advantage-aware policy from offline RL datasets containing mixed-quality data. The proposed method first trains a CVAE over the actions and optimizes the advantage-aware policy towards high advantage values over the learned action embeddings. The experiments show that their method can outperform baselines such as CQL and LAPO.

**Strengths:**

1. The paper is easy to understand.
2. The experiments in the paper demonstrate the effectiveness of A2PO. The agent can learn to sample different actions based on the given advantages and can achieve good performance when given large advantages.

**Weaknesses:**

1. A comparison with diffusion-based methods is lacking. Figure 2 shows that one feature of the proposed method is its ability to model multi-modal actions conditioned on the given state. Recently, diffusion models have become a popular approach for this purpose. How does your method compare to diffusion-based methods? Here are some related papers:
  - Diffusion Policies as an Expressive Policy Class for Offline Reinforcement Learning
  - Planning with Diffusion for Flexible Behavior Synthesis
  - IDQL: Implicit Q-Learning as an Actor-Critic Method with Diffusion Policies
2. As shown in Figure 45, the CVAE over the action space is sensitive to the number of training steps. How can we determine the optimal number of training steps before training an offline RL agent? Is this stage time-consuming?

**Questions:**

Is it possible to use this algorithm in various robot manipulation environments, such as Meta-World and ManiSkill? Additionally, is this stage time-consuming?

---

> ### Author Response · Authors · 2023-11-19
> **Author Response (Part 1/2)**
>
> We are glad that the reviewer appreciates our attempt, and sincerely thank the reviewer for the constructive comments. Below we address the main points raised in the review. We will update the additional experimental results in the revision later.
>
> **[Weakness 1]: compare to the diffusion-based methods**
>
> Thanks for the suggestion. We have additionally conducted experiments of diffusion-based offline RL methods, including Difussion-QL [1] and IDQL [2]. The results in Table R1 show that our A2PO exhibits performance on par with these more heavyweight diffusion-based methods. Notably, diffusion-based methods often perform poorly when there is a significant difference in data quality within the mixed-quality dataset. This observation underscores the effectiveness of our advantage-aware mechanism, which allows our lightweight CVAE model to capture the multi-modal characteristics from the offline dataset more effectively compared to the heavyweight diffusion models.
>
>
> While the diffusion-based methods solely rely on the diffusion models to implicitly learn the distribution of offline data, our advantage-aware mechanism explicitly disentangles the action distributions of intertwined behavior policies by modeling the advantage values of all training data as conditional variables. It is noted that these two approaches are orthogonal. This implies that our advantage-aware mechanism can be effectively integrated with the diffusion-based methods by introducing the conditional diffusion models [3], thereby enhancing their capacity to process multi-modal data. This is an interesting direction for future work.
>
>
> Table R1. Test returns of our proposed A2PO and diffusion-based baselines. *Italics* indicates that the results are obtained from the original paper, while the other results are evaluated over 3 random seeds based on the corresponding official code.
>
> | Environment                                 | Diffusion-QL         | IDQL              | A2PO                  |
> | ----------------------------------- | -------------------- | ----------------- | --------------------- |
> | halfcheetah-medium-v2               | ***51.1***           | *51*              | 47.09 $\pm$ 0.17      |
> | hopper-medium-v2                    | ***90.5***           | *65.4*            | 80.29 $\pm$ 3.95      |
> | walker2d-medium-v2                  | ***87.0***           | *82.5*            | 84.88 $\pm$ 0.23      |
> | halfcheetah-medium-replay-v2        | ***47.8***           | *45.9*            | 44.74 $\pm$ 0.22      |
> | hopper-medium-replay-v2             | *95.5*               | *92.1*            | **101.59 $\pm$ 1.25** |
> | walker2d-medium-replay-v2           | ***101.3***          | *85.1*            | 82.82 $\pm$ 1.70      |
> | halfcheetah-medium-expert-v2        | ***96.8***           | *95.9*            | 95.61 $\pm$ 0.54      |
> | hopper-medium-expert-v2             | ***111.1***          | *108.6*           | 105.44 $\pm$ 0.56     |
> | walker2d-medium-expert-v2           | *110.1*              | ***112.7***       | 112.13 $\pm$ 0.24     |
> | halfcheetah-random-expert-v2        | 86.07 $\pm$ 1.49     | 32.55 $\pm$ 2.94  | **90.32 $\pm$ 1.63**  |
> | hopper-random-expert-v2             | 101.96 $\pm$ 7.03    | 19.73 $\pm$ 13.80 | **105.19 $\pm$ 4.54** |
> | walker2d-random-expert-v2           | 56.33 $\pm$ 31.44    | 0.18 $\pm$ 0.08   | **91.96 $\pm$ 10.98** |
> | halfcheetah-random-medium-v2        | **48.43 $\pm$ 0.32** | 6.26 $\pm$ 1.18   | 45.20 $\pm$ 0.21      |
> | hopper-random-medium-v2             | 6.93 $\pm$ 0.75      | 2.78 $\pm$ 2.01   | **7.14 $\pm$ 0.35**   |
> | walker2d-random-medium-v2           | 3.27 $\pm$ 2.35      | 3.82 $\pm$ 3.21   | **75.80 $\pm$ 2.12**  |
> | halfcheetah-random-medium-expert-v2 | 81.15 $\pm$ 7.21     | 36.24 $\pm$ 13.27 | **90.58 $\pm$ 1.44**  |
> | hopper-random-medium-expert-v2      | 70.09 $\pm$ 2.05     | 6.17 $\pm$ 3.31   | **107.84 $\pm$ 0.42** |
> | walker2d-random-medium-expert-v2    | 56.56 $\pm$ 23.02    | 18.55 $\pm$ 8.13  | **97.71 $\pm$ 6.74**  |
> | **Total**  | 1301.99  | 865.48  | **1466.33**  |
>
> **[Weakness 2]: determine the number of CVAE training steps**
>
> Sorry for the confusion. The number of training steps is a crucial hyperparameter for deep learning models, including our CVAE. A lower number of training steps may result in non-convergence of the model, whereas a higher number can lead to overfitting. In our implementation, we conducted a hyperparameter search on the CVAE training step, denoted as $K$, over $1\times10^5$, $2\times10^5$, $4\times10^5$, and $1\times10^6$. As shown in Table S4 of our original paper, except for the specific requirements in the antmaze and maze2d environments, a value of $K=2*10^5$ proves to be sufficient for all other scenarios. This highlights the robustness of the CVAE training step across various tasks and datasets, without necessitating excessive time expenditure.

---

> ### Author Response · Authors · 2023-11-19
> **Author Response (Part 2/2)**
>
> **[Question 1]: more experiments on the robot manipulation tasks**
>
> Thanks for the constructive comments. Due to the time limitation, we have additionally conducted experiments on the robot manipulation tasks using Adroit and FrankaKitchen instead of Meta-World and ManiSkill, following the previous offline RL works [4,5]. The results in Table R2 indicate that our A2PO method can achieve comparable performance to the state-of-the-art LAPO [5] and SPOT [6], as well as the other baselines.
>
>
>
> Table R2. Test returns of our proposed A2PO and baselines in the robot manipulation tasks of Adroit and FrankaKitchen.
>
> | Environment         | BC     | BCQ    | CQL       | IQL        | TD3BC            | SPOT              | BPPO              | LAPO                  | A2PO                  |
> | ------------------- | ------ | ------ | --------- | ---------- | ---------------- | ----------------- | ----------------- | --------------------- | --------------------- |
> | kitchen-complete-v0 | *33.8* | *8.1*  | *43.8*    | ***62.5*** | 0.83 $\pm$ 1.18  | 41.67 $\pm$ 11.40 | 0.00 $\pm$ 0.00   | 50.83 $\pm$ 10.27     | 60.00 $\pm$ 2.04      |
> | kitchen-partial-v0  | *33.8* | *18.9* | *49.8*    | *46.3*     | 0.00 $\pm$ 0.00  | 0.00 $\pm$ 0.00   | 16.67 $\pm$ 2.36  | **58.75 $\pm$ 16.25** | 48.33 $\pm$ 4.08      |
> | kitchen-mixed-v0    | *47.5* | *10.6* | *51.0*    | *51.0*     | 0.83 $\pm$ 1.18  | 0.00 $\pm$ 0.00   | 0.00 $\pm$ 0.00   | 52.33 $\pm$ 6.27      | **53.33 $\pm$ 2.36**  |
> | pen-human-v1        | *34.4* | *12.3* | *37.5*    | ***71.5*** | -3.69 $\pm$ 0.38 | 32.70 $\pm$ 11.40 | 25.77 $\pm$ 19.25 | 68.06 $\pm$ 18.01     | 68.94 $\pm$ 5.90      |
> | hammer-human-v1     | *1.5*  | *1.2*  | ***4.4*** | *1.4*      | 0.71 $\pm$ 0.25  | 1.99 $\pm$ 0.07   | 0.50 $\pm$ 0.35   | 1.12 $\pm$ 0.29       | 1.84 $\pm$ 0.35       |
> | door-human-v1       | *0.5*  | *0.4*  | ***9.9*** | *4.3*      | -0.33 $\pm$ 0.01 | -0.33 $\pm$ 0.01  | -0.02 $\pm$ 0.03  | 6.07 $\pm$ 4.60       | 8.51 $\pm$ 3.66       |
> | relocate-human-v1   | *0.0*  | *0.0*  | *0.2*     | *0.1*      | -0.30 $\pm$ 0.01 | -0.07 $\pm$ 0.13  | -0.08 $\pm$ 0.08  | 0.04 $\pm$ 0.01       | **0.49 $\pm$ 0.58**   |
> | pen-cloned-v1       | *56.9* | *28.0* | *39.2*    | *37.3*     | 1.74 $\pm$ 2.90  | 2.54 $\pm$ 7.96   | 21.77 $\pm$ 5.88  | 55.84 $\pm$ 20.29     | **84.80 $\pm$ 21.43** |
> | hammer-cloned-v1    | *0.8*  | *0.4*  | ***2.1*** | ***2.1***  | 0.26 $\pm$ 0.02  | 0.46 $\pm$ 0.03   | 0.25 $\pm$ 0.22   | 0.84 $\pm$ 0.35       | 0.43 $\pm$ 0.15       |
> | door-cloned-v1      | *-0.1* | *0.0*  | *0.4*     | ***1.6***  | -0.35 $\pm$ 0.02 | -0.35 $\pm$ 0.02  | -0.04 $\pm$ 0.05  | 0.22 $\pm$ 0.40       | 0.31 $\pm$ 0.04       |
> | relocate-cloned-v1  | -0.1   | *-0.2* | -0.1      | *-0.2*     | -0.31 $\pm$ 0.01 | -0.30 $\pm$ 0.01  | -0.14 $\pm$ 0.11  | -0.05 $\pm$ 0.07      | **0.00 $\pm$ 0.04**   |
> | **Total**  | 209.00   | 79.70 | 238.20  | 277.90   | -0.61 | 78.31  | 64.68  | 294.05  | **326.98**  |
>
>
>
>
> **References**
>
> [1] Wang, Z. et al. Diffusion policies as an expressive policy class for offline reinforcement learning. ICLR 2023.
>
> [2] Hansen-Estruch, P. et al. Idql: Implicit q-learning as an actor-critic method with diffusion policies. arXiv 2023.
>
> [3] Ho, J., & Salimans, T. Classifier-free diffusion guidance. arXiv 2022.
>
> [4] Fu, J. et al. D4RL: Datasets for Deep Data-Driven Reinforcement Learning. arXiv 2020.
>
> [5] Chen, X. et al. LAPO: Latent-Variable Advantage-Weighted Policy Optimization for Offline Reinforcement Learning. NeurIPS 2022.
>
> [6] Wu, J. et al. Supported Policy Optimization for Offline Reinforcement Learning. NeurIPS 2022.

---

> > ### Comment · Reviewer_KGRx · 2023-11-20
> > **Thanks for the response**
> >
> > Thank you for the clarification and experiments. However, I still have doubts regarding the comparison with diffusion-based offline RL algorithms.
> >
> > You mention that diffusion-based methods falter with mixed-quality datasets, but diffusion models are actually used for capturing the multi-modal demonstration distribution. Methods like Diffusion-QL show effectiveness with medium-expert data, at times surpassing your method. Could you explain why they are less effective on random-expert data?

---

> > > ### Author Response · Authors · 2023-11-22
> > > **Author Response**
> > >
> > > Thanks for the response!  We will explain the performance differences between our A2PO and Diffusion-QL in these two datasets from two key aspects:
> > >
> > > (1) Policy Formulation. The core distinction of our A2PO compared to Diffusion-QL does not lie in selecting a conditional model but in introducing a novel policy formulation. Specifically, we utilize $\pi(a|s,\xi)$ with the advantage value $\xi$ to represent the agent policy, while Diffusion-QL utilizes $\pi(a|s)$ to represent the agent policy. Diffusion-QL introduces a conditional diffusion with only the state as the condition to **implicitly** capture the multimodal action distribution, overlooking the intrinsic quality of different actions. However, our A2PO is conditioned on both the state and the advantage value. This advantage-aware policy **explicitly** leverages the advantage value to disentangle the action distributions of intertwined behavior policies, ensuring that the quality of different actions is positively correlated with the advantage value.  It is important to emphasize that the primary efficacy of A2PO stems from this advantage-aware policy, rather than the specific conditional model (be it CVAE or Diffusion).
> > >
> > >
> > > (2) Agent Optimization. The agent optimization of our A2PO and Diffusion-QL contains two main parts, including policy improvement and policy regularization. For our advantage-aware policy, the policy improvement term utilizes $\xi^*=1$ to explicitly learn agent policy from high-quality action distribution, while the policy regularization term imposes constraints on the advantage-aware policy with estimated $\xi$ of collected samples to ensure the agent policy selects in-sample actions.  Therefore, the suboptimal samples with low advantage condition $\xi$ will not disrupt the optimization of optimal policy. But for Diffusion-QL, its policy regularization term encourages the policy to sample actions with the support of collected samples without identifying the quality of different actions, resulting in unreliable policy improvement.
> > >
> > > For the medium-expert dataset, the quality of samples from both sources is relatively similar. Thus, Diffusion-QL can rely on a high-expressiveness foundation model to achieve promising results, while the disentanglement benefit brought by our advantage-aware policy is not obvious. However, the random-expert datasets highlight the issue of substantial gaps between behavior policies. Thus, the Diffusion-QL imposing incorrect constraints on the low-quality samples will significantly disrupt the agent policy optimization for high-quality actions. In contrast, our advantage-aware policy separates the samples and only considers high-quality data for policy constraints, ensuring a stable and superior agent policy.
> > >
> > >
> > > ---
> > >
> > > **We appreciate your time and effort in reviewing our work. We have taken your comments into careful consideration and conducted extensive experiments to address the concerns you raised. As suggested, we have additionally conducted comparison experiments with more SOTA baselines on more robot tasks. Moreover, we have also explained our hyperparameters setting and the different contributions compared with diffusion-based methods. Please let us know if you have other questions or comments.**
> > >
> > > **We sincerely look forward to your reevaluation of our work and would very appreciate it if you could raise your score to boost our chance of more exposure to the community. Thank you very much!**

---

> > > > ### Comment · Reviewer_KGRx · 2023-11-22
> > > > **Thanks for the explanation**
> > > >
> > > > Thanks for the efforts! My concerns are fully addressed. I vote for the acceptance! **I have raised my score from 5 to 6.** My feeling is that your method may get better trade-off between modeling the demonstration distribution and policy optimization. Also, I encourage the authors to further explore this direction with diffusion models.

---

> ### Author Response · Authors · 2023-11-23
> **Thanks for the positive support!**
>
> Dear Reviewer,
>
> We sincerely appreciate your positive feedback, the time and effort you have put into reviewing our modifications and responses!  It is immensely gratifying to learn that our revisions have successfully addressed your concerns. Your suggestion to further explore this direction using diffusion models is invaluable, and we are enthusiastic about incorporating this perspective into our future research.
>
> Thank you once again for your constructive feedback and support.
>
> Best,
>
> Authors

---

### Author Response · Authors · 2023-11-22
**Thank you to all reviewers.**

Thank you again to all reviewers for the time and effort you spent reviewing our paper. We have tried to incorporate all your feedback and suggestions in our updated manuscript. Here is a list of the changes:

- Updated Section 2 with a brief discussion about reward-conditioned supervised learning as suggested by reviewers TZv3.
- Updated Section 4 and Appendix B with more details about TD3BC as suggested by reviewer HeZ6.
- We moved the pseudo algorithm 1 to the Appendix A Section with more detail of A2PO  as suggested by reviewer Mmfe.
- Added Appendix I  evaluating our A2PO and other baselines on more robot manipulation tasks as suggested by reviewers KGRx and TZv3.
- Added Appendix J  where we evaluated our A2PO,  diffusion-based baselines, representation learning baselines, and more lightweight baselines as suggested by reviewers KGRx, Mmfe, and TZv3.
- Added Appendix K  comparing our A2PO without BC regularization term with the LAPO method as suggested by reviewer R6oW.
- Added Appendix L  where we tested A2PO under datasets containing different proportions of single-quality samples.  as suggested by reviewer Mmfe.
- Added many smaller edits throughout the paper that are described in our individual responses to the reviewers.

**We hope these updates address your concerns. If so, we ask that you consider raising your score, and please let us know any changes that would be helpful for a potential camera-ready version.** Thank you again and we hope you will consider our paper an important contribution to the ICLR community.

---

### Meta-Review · Area_Chair_iBCZ · 2023-12-05

**Metareview:**

This paper proposes a new method called A2PO that aims to learn an advantage conditioned policy with offline RL from mixed quality data. This works by iterating between policy optimization and using a CVAE to disentangle policy quality within the latent space of the actor.

While this paper has received mixed reviews, I recommend that it be rejected for the following reasons:
- There is no strong support for accepting this paper from the reviewers.
- There are substantial concerns about the proposed methods and the experimental evaluations: (a) the proposed method makes the architecture heavier with the CVAE, (b) the experimental data seems unrealistic with the inclusion of suboptimal random data in the mix, (c) the method could be better compared to diffusion models.

**Justification For Why Not Higher Score:**

While this paper has received mixed reviews, I recommend that it be rejected for the following reasons:
- There is no strong support for accepting this paper from the reviewers.
- There are substantial concerns about the proposed methods and the experimental evaluations: (a) the proposed method makes the architecture heavier with the CVAE, (b) the experimental data seems unrealistic with the inclusion of suboptimal random data in the mix, (c) the method could be better compared to diffusion models.

**Justification For Why Not Lower Score:**

N/A

---

### Decision · Program_Chairs · 2024-01-16

Reject